# Pathways for horizontal gene transfer in bacteria revealed by a global map of their plasmids

Santiago Redondo-Salvo [1,5], Raúl Fernández-López [1,5], Raúl Ruiz [1], Luis Vielva [2], María de Toro[3], Eduardo P. C. Rocha [4], M. Pilar Garcillán-Barcia [1] & Fernando de la Cruz[1✉]

Plasmids can mediate horizontal gene transfer of antibiotic resistance, virulence genes, and other adaptive factors across bacterial populations. Here, we analyze genomic composition and pairwise sequence identity for over 10,000 reference plasmids to obtain a global map of the prokaryotic plasmidome. Plasmids in this map organize into discrete clusters, which we call plasmid taxonomic units (PTUs), with high average nucleotide identity between its members. We identify 83 PTUs in the order *Enterobacterales*, 28 of them corresponding to previously described archetypes. Furthermore, we develop an automated algorithm for PTU identification, and validate its performance using stochastic blockmodeling. The algorithm reveals a total of 276 PTUs in the bacterial domain. Each PTU exhibits a characteristic host distribution, organized into a six-grade scale (I–VI), ranging from plasmids restricted to a single host species (grade I) to plasmids able to colonize species from different phyla (grade VI). More than 60% of the plasmids in the global map are in groups with host ranges beyond the species barrier.

[1] Instituto de Biomedicina y Biotecnología de Cantabria (IBBTEC), Universidad de Cantabria-CSIC, C/Albert Einstein 22, 39011 Santander, Spain. [2] Departamento de Ingeniería de las Comunicaciones, Universidad de Cantabria, Santander, Spain. [3] CIBIR, Centro de Investigación Biomédica de La Rioja, Logroño, Spain. [4] Microbial Evolutionary Genomics, Institut Pasteur, CNRS, UMR3525 Paris, France. [5] These authors contributed equally: Santiago Redondo-Salvo, Raúl Fernández-López. ✉email: delacruz@unican.es

Prokaryotes lack sexual reproduction, but undergo extensive horizontal gene transfer (HGT), acquiring DNA from individuals other than their immediate ancestors[1]. HGT is a fundamental force driving bacterial evolution. The existence of recognizable clades in prokaryotic lineages implies that, although rampant[2], HGT is not strong enough to blur the traces of vertical descent[1,3,4]. This indicates that there are factors limiting HGT, yet the nature of these factors is a matter of debate[5]. Metagenomics revealed the existence of "habitat-specific gene pools", suggesting that transmission is more frequent in species coresiding in the same ecological niche[6–8]. Genomic analysis, however, showed that HGT decreases with phylogenetic distance, pointing out to higher genetic relatedness as the key factor limiting gene transfer[2,9–11]. Determining the relevance of these boundaries is essential to understand the impact of HGT on the evolution and persistence of bacterial species[1,12]. In bacteria, HGT is mostly the byproduct of the infectious spread of mobile genetic elements (MGEs) such as bacteriophages, integrative and conjugative elements (ICEs), and plasmids[13–15]. Limits for MGE propagation constitute foremost barriers for HGT in microbes. Bacteriophage propagation is limited mainly by host genetic similarity, rather than ecological opportunity, so transduction occurs mainly within the species boundary[16]. Plasmids and ICEs, however, are known to cross the interspecies barrier[13,14]. Their promiscuity may thus determine the scope and frequency of HGT between different clades[17].

Many efforts have been invested in determining the host range of conjugative elements, both experimentally and by in silico analyses[18–27]. A common problem arising in these studies resides in the difficulty of generalizing properties observed for a given plasmid to "similar" ones. The habitat range of a given organism can be determined because it is possible to assign individuals to the same functional group (the species). Similarly, determining the host range of a certain plasmid requires ways to establish which plasmids can be considered equivalent. This is not a trivial task. Historically, plasmids were classified in incompatibility (Inc) groups, defined by the inability of their members to coreside in the same cell[28,29]. Inc groups loosely reflect higher genetic relatedness[28], and in some well-studied cases, like IncP-1 and IncW groups, there is evidence of a basic genomic backbone characteristic of the group[30–32]. However, plasmids can be incompatible by a variety of mechanisms, and compatibility is not always proportional to genetic distance. Members of the same Inc group often exhibit important genetic differences, and not all groups entail similar levels of sequence conservation[30–32]. Moreover, plasmids from different Inc groups are known to recombine[25,26,33], creating reticulated phylogenies in which different parts of the genome exhibit alternate evolutionary trajectories[26,31]. For conjugative plasmids, it is possible to use the conjugation genes as common phylogenetic tracers[34–37] and even derive a taxonomy upon them[36]. However, it is unclear whether, at the distal end of this phylogeny, there is anything similar to a "molecular species": a group of genetically coherent genomes that evolve together.

Preferential recombination and ecological cohesiveness are major forces maintaining the genomic coherence of bacterial species[1,4,6,12,38]. Here we investigate whether such forces also operate in plasmid populations. For this purpose, we analyze the genomes of over 10,000 reference plasmids, obtaining a global map of the prokaryotic plasmidome. We show that plasmid sequences form discrete clusters, which we call PTUs (plasmid taxonomic units). This indicates that plasmids, like their hosts, form coherent genomic groups, similar to molecular species. PTUs thus represent a natural classification scheme for plasmids. An exploration of their diversity reveals that PTUs sometimes correlate to classical Inc groups, although most PTUs do not have a clear Inc counterpart. Results also unveil a repertoire of largely unexplored plasmid archetypes. Most nonmobilizable PTUs appear circumscribed to a single bacterial species, while those encoding a conjugation apparatus spread across entire bacterial families. Only a fraction (<10%) of the PTUs identified appear in species from different bacterial orders. However, this small fraction of highly promiscuous plasmids, combined with a large number of PTUs able to colonize entire bacterial families, form a vast network for genetic exchanges in bacteria.

## Results

**Plasmid propagation is limited by phylogenetic barriers**. To determine the boundaries for HGT in prokaryotes, we first studied the distribution of the plasmid-encoded genome in *Bacteria* and *Archaea*. For this purpose, we employed AcCNET, a bioinformatics pipeline intended to analyze bacterial accessory genomes[39]. We analyzed 10,634 bacterial and archaeal plasmids present in the RefSeq84 database. AcCNET extracted the plasmid-encoded proteome and organized it in 218,236 homologous protein clusters (HPCs). Then, a bipartite network was created, containing nodes corresponding to plasmid genomes and nodes corresponding to HPCs. When a plasmid genome encoded a member of a given HPC, both nodes were linked by an edge. Edges and nodes are organized using the ForceAtlas2 algorithm, thus the network self-organizes according to overall genomic similarity[39]. The overall network for the prokaryotic plasmidome contained 890,006 edges, as shown in Fig. 1a, b. In this network, plasmid nodes grouped in a number of dense clusters, corresponding to sets of plasmids with high overall similarity in their proteome composition. When we colored the plasmid nodes according to the phylum of their host, we observed that they separated in different territories, corresponding to major bacterial phyla, including *Proteobacteria*, *Firmicutes*, *Actinobacteria*, *Spirochaetes*, *Cyanobacteria* and *Archaea* (Fig. 1a). Clustering along phylogenetic lines was also observed when we descended one taxonomic level and looked at the α, β, and γ subdivisions in the phylum *Proteobacteria* (Fig. 1c). An equivalent situation occurred when we analyzed the orders of class γ-*Proteobacteria* (Fig. 1d). However, this trend dissipated when we looked within the *Enterobacteriaceae* family (Fig. 1e, f), indicating that the plasmid-encoded genome is widely shared among different genera in this family. To test if this trend was observed in the entire prokaryotic plasmidome, we measured the fraction of HPCs that included plasmids from different taxa, at different taxonomic levels. As shown in Fig. 2a, this fraction decreased with phylogenetic distance, especially above the order rank. Similar results were observed when we measured the fraction of total trajectories linking plasmids from different clades (Supplementary Fig. 1a), and when we measured the fraction of plasmids per clade exhibiting a shared cluster (Supplementary Fig. 1b). Altogether, these metrics indicated a gradient of plasmid sharing along taxonomic boundaries, suggesting that host similarity acts as a major constraint for the propagation of plasmid-encoded genes.

**Plasmids organize in genomic clusters**. The above results suggested that the gene repertoire associated with plasmids and shared by different bacteria was constrained by host phylogeny. Yet, to determine the distribution range of the plasmids themselves, we needed a systematic measure definition of which plasmids can be considered genetically equivalent. Following recent work on bacterial species[4,40], we examined average nucleotide identities (ANI). In bacterial chromosomes, pairwise ANI values display a bimodal distribution, with members of the same species exhibiting ANI > 95%, while members of different species produce results below 84%[4,40]. This fact can be used to

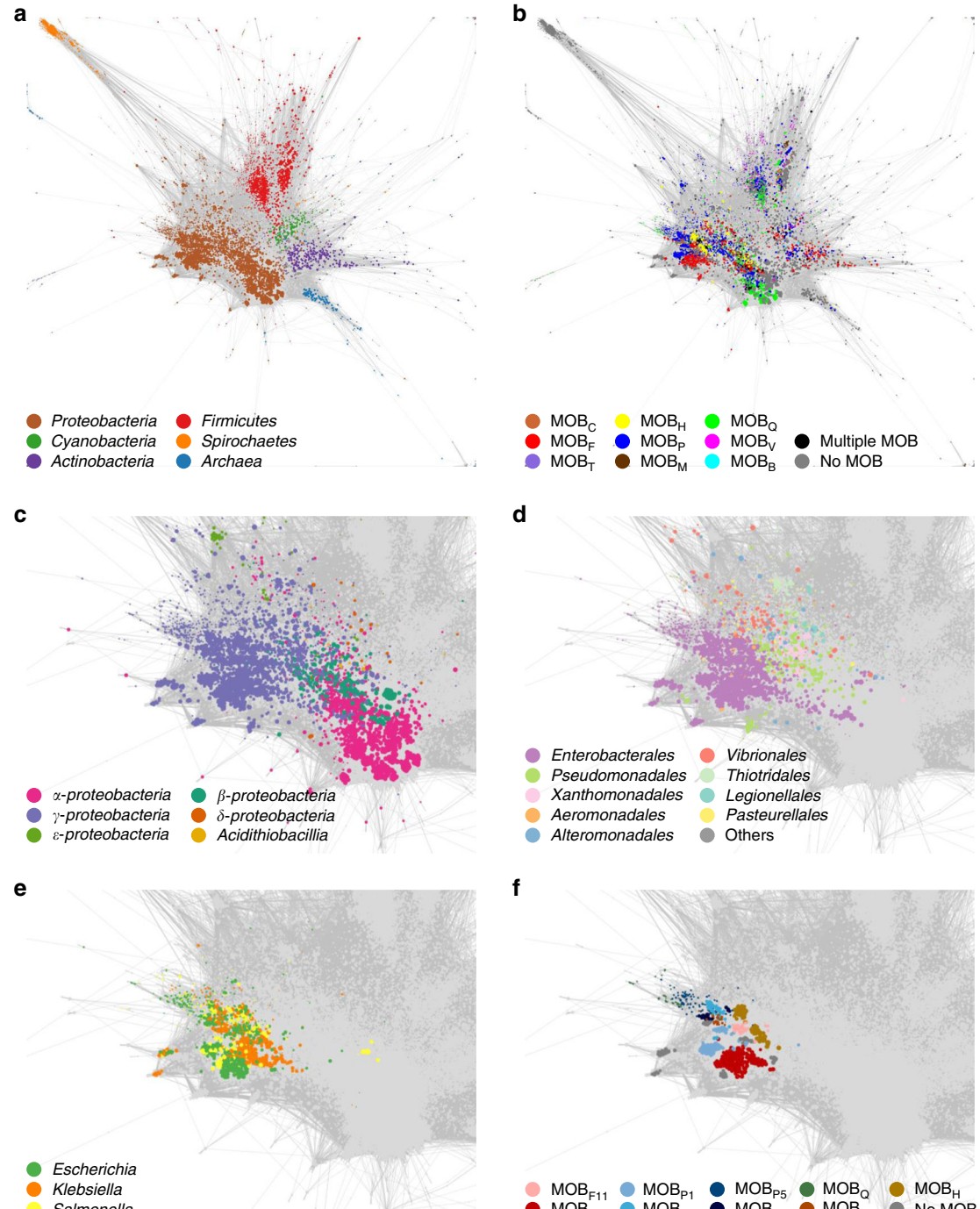

**Fig. 1 Clustering of plasmid genomes along taxonomic boundaries.** All graphs represent AcCNET bipartite networks, with nodes representing plasmid genomes (P) or homologous protein clusters (HPC), extracted from (P) as indicated in Methods. Whenever a P node encodes a member of a given HPC, an edge between both nodes is drawn. Edge lengths depend on the similarity to the most representative member of the cluster, as defined by kClust. **a**, **c–e** panels show clustering by the host. Plasmid nodes were colored, as indicated in the respective color codes, by host phylum (**a**), host class within the *Proteobacteria* phylum (**c**), order-level within γ-*proteobacteria* (**d**), or genus-level, for the *E. coli–Salmonella–Klebsiella* group (**e**). **b**, **f** panels show clustering by MOB types in the overall plasmidome network (**b**) or in the family *Enterobacteriaceae* (**f**). Source data are provided as a Source Data file.

attribute individuals to particular species based solely on genomic comparisons[4,40,41]. Applying the same approach to plasmids, however, is not straightforward. On the one hand, there is not a universal core of genes shared by all plasmids, thus comparisons of two plasmid backbones may yield ANI scores of 0. On the other hand, transposons, insertion sequences, and other MGEs often hop on plasmid genomes. This causes that otherwise unrelated plasmids show certain fragments of their genomes with

high ANI values. Because of these two reasons, the pairwise ANI between two plasmids may be biased by the presence of common MGEs harbored in their genomes. To correct for this bias, we needed some way of accounting for the percentage of the plasmid genome that exhibits detectable ANI. This can be achieved by measuring the alignment fraction (AF), a score of the percentage of two DNA molecules that can be unambiguously aligned above a certain threshold[40]. However, measuring AF is computationally

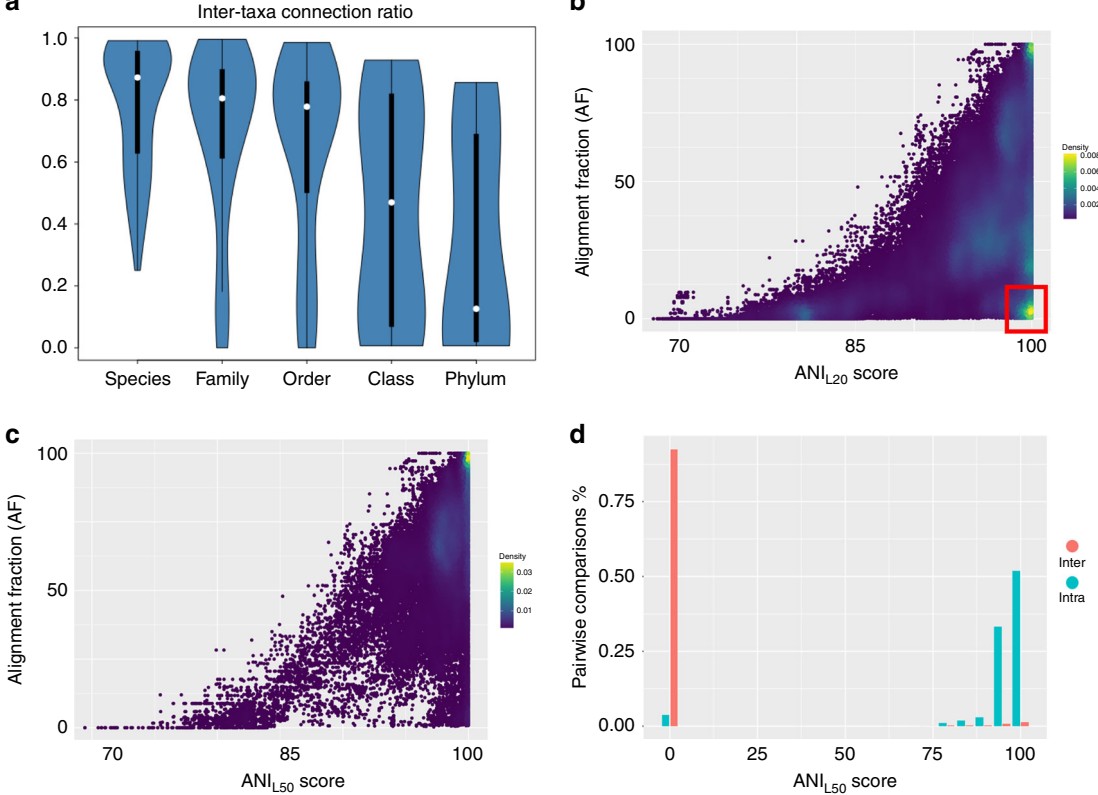

**Fig. 2 Measuring plasmid similarity by AcCNET and ANI. a** Plasmids of the same taxon tend to be more alike. Violin plots showing the ratio of trajectories between plasmids from different taxa in the AcCNET network of Fig. 1. A trajectory is defined as a connection between a plasmid, a given HPC, and a second plasmid that also contains a member of the HPC. For each plasmid of each taxon considered, the ratio was obtained by dividing the number of trajectories linking plasmids hosted in different taxa (species, families, phyla…) by the total number of trajectories of that plasmid. White dots indicate the median connection ratio of trajectories between plasmids, black bars correspond to the Q1-Q3 interquartile range. **b** Density plot of $ANI_{L20}$ (x-axis) and AF (y-axis) values for pairwise comparisons between plasmids of the order *Enterobacterales*. The color code indicates the relative density of that particular ANI/AF area. Yellow areas correspond thus to more frequent ANI/AF scores. The red square highlights a high number of comparisons with $ANI_{L20} > 90\%$ but AF < 10, corresponding to plasmids sharing only a small fragment of their genome with high homology (e.g., transposons or insertion sequences). Separate comparisons for MOB+ and MOB− plasmids with sizes higher and lower than 40 kb are shown in Supplementary Fig. 3. **c** Density plot of $ANI_{L50}$ (x-axis) versus AF (y-axis). Yellow areas correspond to more frequent ANI/AF scores. As shown in the figure, the L50 threshold on the ANI algorithm eliminates the high ANI/low AF comparisons. **d** Histogram of $ANI_{L50}$ scores obtained in comparisons between plasmids of the same PTU (blue) and between plasmids of different PTUs (red).

intensive, which precludes its general application to the entire prokaryotic plasmidome. For these reasons, we first compared the results of applying the ANI algorithm with minimal length threshold of 20% ($ANI_{L20}$), and the AF algorithm in the subset of plasmids from the order *Enterobacterales*. $ANI_{L20}$ refers to the total length of the shortest plasmid in the comparison. Pairwise $ANI_{L20}$ and AF scores were obtained as described in Methods. Results revealed that 61% of the pairwise comparisons yielded DNA identity below the 70% identity threshold. The plot of AF versus $ANI_{L20}$ values of the remaining is shown in Fig. 2b. As shown, most pairwise comparisons distributed around two areas of high density (yellow areas in Fig. 2b). The first one included plasmid pairs showing >95% both in $ANI_{L20}$ and AF scores, corresponding to plasmid pairs highly similar along most their genomes. The second included comparisons yielding >95% ANI scores in regions covering <10% of the AF, i.e., plasmids that have a small but highly similar homologous region. The high ANI/low AF areas mostly correspond to transposons, insertion sequences, and other MGEs inserted in different plasmid backbones, which are extremely abundant (as an example, Tn3 transposases were found in 12% of the proteobacterial plasmids, Supplementary Data 1 and Supplementary Table 1). In order to filter spurious ANI scores due to MGEs, we calculated $ANI_{L50}$ (ANI with a 50%

length threshold). A comparison of $ANI_{L50}$ against AF scores in the *Enterobacterales* plasmidome (Fig. 2c), revealed that this length threshold was well suited to discard high ANI values with small AF. $ANI_{L50}$, which is less computationally intensive than the AF algorithm, will be hereafter used in this work to infer overall plasmid sequence identity.

The absence of pairwise comparisons with intermediate ANI suggested, as in bacterial chromosomes[4], that plasmids clustered in discrete groups, instead of spreading over a continuous genetic landscape. To test it, we built a monopartite graph where nodes represented plasmid genomes, and edges corresponded to ANI scores (Fig. 3a, b). In this network, which provides a global representation of plasmid diversity, plasmids with no significant similarity to other members of the set appear as singletons, circumscribed to the periphery of the graph. The remaining plasmidome organized in different paracliques: sets of nodes showing numerous mutual connections. We tested whether the overall network structure depended on the length threshold of the ANI score, by varying it from $ANI_{L20}$ to $ANI_{L70}$ in the subset of plasmids from *E. coli* (Supplementary Fig. 4). As shown in the figure, changing the length threshold does not alter the overall clustering tendency, but increases or decreases the density of connections between paracliques. Judging from the AF vs ANI

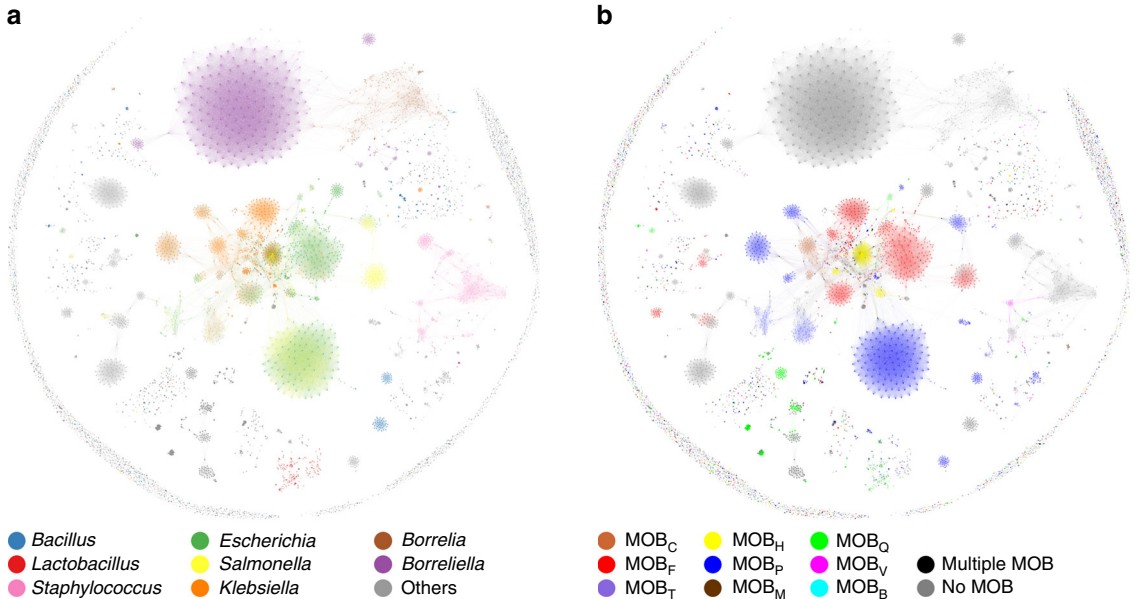

**Fig. 3 ANI$_{L50}$ similarity network of the bacterial plasmidome.** Similarity networks of the RefSeq84 prokaryotic plasmidome obtained using the ANI$_{L50}$ algorithm as described in Methods. Nodes, corresponding to plasmid genomes, are colored according to their cognate host taxonomy (**a**) or MOB class, as defined by MOBscan (**b**). Source data are provided as a Source Data file.

plot of Fig. 2b, we opted for ANI$_{L50}$ threshold to include only ANI scores pertaining to the majority of the plasmid backbone.

**Plasmids clusters contain a common genomic backbone.** The ANI$_{L50}$ network of prokaryotic plasmids shown in Fig. 3 allowed us to compute a series of metrics to quantitatively identify plasmid clusters. Clusters were defined upon three criteria. First, we considered only paracliques formed by at least four plasmids, in order to have sufficient sampling of the genomic-sequence space. Second, the intracluster density, calculated as described in Methods, was at least 25%, indicating that every member of the paraclique was connected to at least one out of four members of the cluster. Finally, the intra to interdensity ratio was set to >500, indicating that members of a given cluster were at least 500 times more likely to show a connection with members of the same paraclique, than with plasmids outside their group. When we applied these criteria to plasmids in the order *Enterobacterales*, 83 plasmid groups were found. Large networks may display artifactual random clustering masquerading as a bona fide network structure. To check that our groups represented true communities rather than stochastic clusters, we used Bayesian Stochastic Blockmodel (SBM), a technique that allows the statistical inference of communities in graphs[42,43]. Details on the implementation and validation of SMBs for plasmid cluster detection are described in Supplementary Methods. SBM results showed statistical support for 85% of the plasmid clusters identified in the order *Enterobacterales*, which included a total of 2535 plasmids. From these, 1770 were included in 83 PTUs in *Enterobacterales*. These clusters represented groups of plasmids that, as judged from their ANI scores, present a high level of DNA similarity among them. The distribution of pairwise ANI$_{L50}$ scores (Fig. 2d) revealed that more than 63% of the comparisons between members of the same group showed ANI$_{L50}$ >90%. In contrast, 99.75% of the comparisons between members of different groups showed ANI$_{L50}$ = 0. Results thus indicate that these clusters represent plasmid groups with a common, coherent genome.

Mobilizable and conjugative plasmids can be classified according to their mobilization functions (MOB) as described in Methods. When we analyzed their MOB types, we discovered

that conjugative groups were characterized by a particular relaxase (the protein that recognizes the plasmid origin of transfer, whose gene is a constituent of MOB) and a conserved transfer machinery (Fig. 4b, c). These clusters thus fulfilled the criteria of similarity (members are more alike between them than with individuals outside the cluster) and contain phylogenetic markers that can be used to trace common descent. These two conditions are employed to classify bacterial species[44], and were also proposed for plasmids[36], thus we hereafter refer to these clusters as Plasmid Taxonomic Units (PTUs).

**PTUs and their correlation to classical Inc groups.** A number of PTUs showed a correlation to classical incompatibility groups, reinforcing the idea that certain Inc groups have a phylogenetic character. Some of them had a direct correlation, such as IncL/M, IncC, and IncB/O/K/Z plasmids, each represented by a single PTU. In other cases, however, the correlation was between a PTU and a given subdivision of a classical Inc group, defined in the literature by certain phenotypic traits or, more frequently, by multilocus sequence typing[45–48]. For example, we did not find a PTU corresponding to IncX plasmids, but rather identified three PTUs corresponding each to a known subdivision of the IncX (PTU-X1, -X3, and -X4). In these cases, we kept the capital letter of the corresponding subdivision (e.g., PTU-I1 plasmids). The most extreme case of an Inc group corresponding to several different PTUs was the IncF. This was not surprising, as it has been known for long that the IncF complex is a broad category that includes many different plasmid archetypes[32]. IncF plasmids partitioned in eight different PTUs. Many of them corresponded to characteristic pMLST profiles, for example, IncF$_K$ plasmids from *Klebsiella*[49]. Following this naming convention, and given that most IncF groups corresponded to specific bacterial hosts, we named then accordingly (PTU-F$_E$, for *Escherichia*, PTU-F$_S$ for *Salmonella*, PTU-F$_{Sh}$ for *Shigella*, PTU-F$_Y$ for *Yersinia*). IncFV plasmids segregated into a separate cluster (PTU-FV), as expected from a previous analysis[32]. Out of 83 PTUs identified in *Enterobacterales*, 55 did not belong to known Inc groups. Given the lack of a reported IncE group in the literature, the remaining

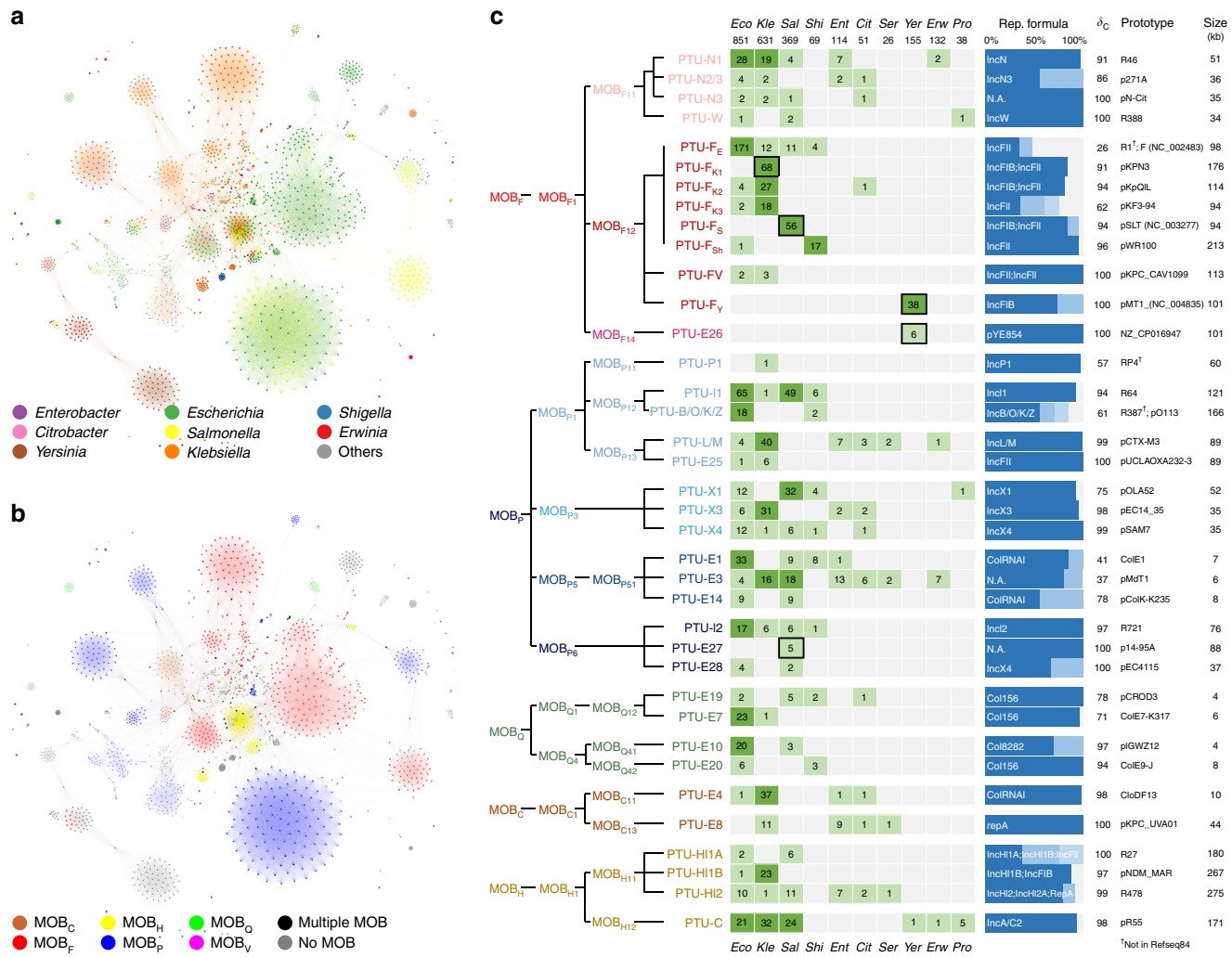

**Fig. 4 Conjugative and mobilizable PTUs of the order *Enterobacterales*.** The figure represents an integrative view of the PID+SBM methods. **a**, **b** ANI network of the *Enterobacterales* plasmidome, with plasmids colored according to their cognate host (**a**) or the MOB type of each plasmid (**b**). Source data are provided as a Source Data file. **c** A list of the most representative mobilizable and conjugative PTUs from *Enterobacterales*, according to their MOB type. Columns on the left indicate the MOB type and name of the PTU. Numbers in the boxes indicate the number of plasmids isolated from the different genera shown in the figure. Dark green squares indicate high plasmid prevalence, with >15 different plasmids retrieved from that genus. Boxes correspond, respectively, to *Escherichia* (*Eco*), *Klebsiella* (*Kle*), *Salmonella* (*Sal*), *Shigella* (*Shi*), *Citrobacter* (*Cit*), *Serratia* (*Ser*), *Yersinia* (*Yer*), *Erwinia* (*Erw*), and *Proteus/Providen*cia (*Pro*). The number below each taxon represents the total number of plasmids from that taxon present in the database. Boxes squared with a black margin represent PTUs exclusively present in that particular host. The blue bar on the right indicates the most prevalent replicon structure, as determined by PlasmidFinder. The rightmost two columns of the figure represent, respectively, the intracluster density, and a prototype plasmid from the group.

groups were named PTU-E (from *Enterobacterales*) followed by a number (PTU-E1 to E55).

Relaxase and replicon typing was performed in all plasmids using, respectively, MOBscan[50] and Plasmid Finder[51]. A representation of the mobilizable and conjugative PTUs, as judged by the presence of a conjugative relaxase, is shown in Fig. 4. Figure 4a is colored according to the host bacteria, while Fig. 4b is colored according to MOB type. A representation of nonmobilizable PTUs is shown in Supplementary Fig. 5. Of the enterobacterial PTUs, 37 showed a characteristic relaxase type. $MOB_F$ and $MOB_P$ were the most frequent, with 13 and 14 representative PTUs, respectively. Each PTU was characterized by a single relaxase type, except for a few PTUs in which mobilization functions were lost in some of their members (PTU-$F_Y$ virulence plasmids in *Yersinia pestis* and PTU-$F_S$ plasmids from *Salmonella*, for example). In contrast, replication functions within a given PTU showed considerable variation (Fig. 4c). An overall 60% of the

enterobacterial PTUs exhibited a characteristic replicon formula (a replicon or combination of replicons present in >90% of group members). The remaining 40% showed several replicon combinations. These results thus indicate that, while the association of PTUs to a particular replicon type is frequent, plasmids maintain their genomic identity despite shuffling replication machineries. Furthermore, replicon types are also not exclusive of a given PTU, since the same replication formula could be found in different PTUs.

**PTUs show a scale of host ranges**. Conjugative and mobilizable plasmids were traditionally categorized into broad and narrow host range, depending on their ability to colonize different bacterial species[34]. When we colored enterobacterial plasmids according to their cognate hosts, we realized that this distinction was unfit to describe the actual host range of PTUs. Some PTUs

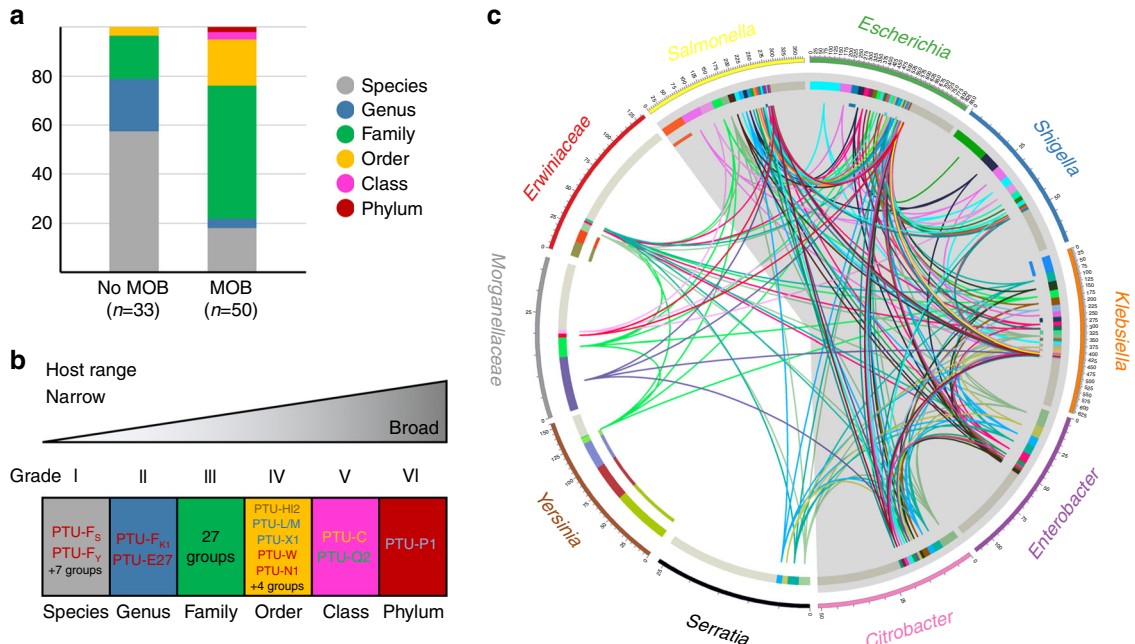

**Fig. 5 PTUs of different host range delineate pathways for HGT in *Enterobacterales*. a** The graph shows the host distribution of different PTUs. The distribution breath is the taxonomic depth between the two most distinct hosts in which that PTU is found. The left column shows percentages for nonmobilizable PTUs, while the right column shows data for mobilizable and conjugative PTUs. **b** Host range scale for mobilizable and conjugative PTUs from *Enterobacterales*, color-coded as in **a**. **c** Chord diagram showing the distribution of PTUs among the bacterial taxa shown in the outer rim of the diagram. Bars below each taxonomic group represent the total number of plasmids present in that taxon. Colored sectors indicate each of the mobilizable/conjugative PTUs defined in Fig. 4b. Inner ring is colored by PTU membership of plasmids. Whenever two bacterial taxa host plasmids from the same PTU, they are linked by an edge. This way, internal chords represent potential routes for plasmid-mediated HGT among different bacterial genera. Members of the *Enterobacteriaceae* family are gray-shadowed.

were circumscribed to a single species, for example, PTU-E26 plasmids from *Yersinia enterocolitica*, while others were present in species from even different phyla, as for PTU-P1 plasmids. In between, we observed a gradient of host breaths. Classifying plasmids according to the highest taxon they distribute in, we obtained a six-grade scale (Fig. 5). As expected, mobilizable and conjugative plasmid groups exhibited broader host distributions than nonmobilizable groups (Fig. 5a). Most nonmobilizable enterobacterial PTUs were circumscribed to Grades I and II, corresponding to species or genera (84% of the total). In contrast, 81% of the mobilizable and conjugative PTUs distributed in Grade III or higher (Fig. 5b and Supplementary Data 2). Grade III was the most frequent among conjugative and mobilizable plasmids (60%), corresponding to PTUs that typically colonized species in the *Enterobacteriaceae* (*Escherichia*, *Klebsiella*, and *Salmonella*). Grade IV was achieved by 9 PTUs (L/M, N1, W, HI2, X1, E3, E8, E12, and E50). These PTUs typically colonized different species in the *Enterobacteriaceae* and some others from *Yersiniaceae*, *Erwiniaceae*, or *Morganellaceae* families. Grade V was achieved by PTU-C plasmids, colonizing *Enterobacteriaceae*, *Erwiniaceae*, and *Morganellaceae*, but also species from orders *Vibrionales*, *Alteromonadales*, and *Aeromonadales*, and PTU-Q2 plasmids, colonizing *Enterobacteriaceae* and *Aeromonadaceae*. Grade VI was reached by PTU-P1 plasmids alone, which distributed throughout γ-*proteobacteria* (including *Klebsiella*, *Pseudomonas*, and *Xanthomonas*), α-*proteobacteria* (*Sphingomonas*), β-*proteobacteria* (*Cupriavidus*, *Comamonas*, *Delftia*), and even *Actinobacteria* (*Mycobacterium*). Although most host species harboring PTU-P1 plasmids belong to the order *Burkholderiales*, plasmids from this group can be found scattered throughout the entire Bacterial kingdom. Previous experimental data showed that

PTU-P1 plasmids exhibit high-host promiscuity[19,20], supporting their role as the PTU with the widest host range[27].

Based on the catalog of PTUs and their respective host ranges, we were able to trace potential pathways of plasmid exchanges between taxa. Mobilizable and conjugative PTUs with the ability to colonize different taxa may serve as vehicles for interspecies HGT. Thus, we analyzed the possible exchange pathways in clinically-relevant species from the order *Enterobacterales* (Fig. 5c). In the chord diagram shown in the figure, two taxa are connected whenever there is a mobilizable/conjugative PTU able to colonize both. As shown in the figure, connections between members of the *Enterobacteriaceae* family are abundant. A total of 17 different PTUs may mediate genetic exchanges between *Escherichia* and *Klebsiella*, while 15 connect *Escherichia* and *Salmonella*. Plasmid sharing between members of the *Enterobacteriaceae* family was more frequent than with members of other families. However, *Escherichia*, *Klebsiella*, and *Salmonella* contain at least one PTU shared with all the taxa shown in the figure. This indicates that these species may directly exchange plasmids with most of the clinically-relevant bacteria of the order *Enterobacterales*.

**PID, an automated algorithm for PTU identification.** So far, our analysis was circumscribed to plasmids from the order *Enterobacterales*. However, due to the size of the prokaryotic plasmid database, we needed an automated method for PTU detection. The goal was to implement an algorithm able to robustly identify paracliques inside the ANI$_{L50}$ network. Common methods based on Voronoi decomposition on a 2D plane, such as K-means, were poorly suited. In our networks, it is only

the topology that depends on ANI values. The 2D configuration is contingent on the ForceAtlas algorithm, which may produce different global arrangements in each execution. For this reason, we implemented a topological approach. Our algorithm, named PID (plasmid identification), divides a graph into its connected components, cliques and paracliques. The algorithm performs this task by iteratively eliminating hubs between connected components, and comparing the result of each step with ideal cliques of the same number of nodes. We validated PID by applying it to the $ANI_{L50}$ network of the *Enterobacterales* plasmidome. All PTUs identified by PID coincided with those defined by calculating clustering densities and SBM, except for the PTU-$F_E$ group. A closer inspection of PTU-$F_E$ revealed that this group presents the lowest intracluster density ($\delta_c = 0.24$, Supplementary Data 3), and is divided into five different clusters (Supplementary Data 4). This suggests that there may be alternate configurations of PTU-F plasmids in *E. coli*, a result already suggested in a previous analysis[32]. Altogether, PID results presented a 98.2% overlap with the previous clustering methods, indicating that a combination of this topological approach with the statistical support yielded by SBMs is suited for automated identification of plasmid clusters.

**A global map of the prokaryotic plasmidome.** When we applied SBM and PID to the $ANI_{L50}$ network of the entire prokaryotic plasmidome, we identified 276 statistically supported PTUs with at least four or more member plasmids, which are shown in Fig. 6a. An interactive map of the prokaryotic plasmidome can be accessed at https://castillo.dicom.unican.es/PlasmidID/, and a complete list of all PTUs identified can be found on Supplementary Data 2. The PTUs identified included a total of 3410 plasmids, 32% of the total, indicating that a majority of plasmids in the database belonged to groups yet to be characterized. Approximately half of the PTUs identified contained a characteristic relaxase. However, the number of MOB+ to MOB− plasmids classified was 2:1, indicating that MOB+ PTUs were more populated than their nonmobilizable counterparts. In terms of host range, as in the case of enterobacterial plasmids, non-mobilizable PTUs exhibited lower host ranges, with 81% of the total belonging to Classes I and II. Mobilizable and conjugative PTUs, on the other hand, exhibited Class III or higher in 52% of the cases. As in *Enterobacterales*, the most frequent host range for nonmobilizable PTUs was Class I (restricted to a single species), while for mobilizable and conjugative PTUs was Class III (distributed among different members of a bacterial family). These numbers were biased by differences in sampling across bacterial taxa. As shown in Fig. 6b, there were only six bacterial orders in which we could identify at least 10 PTUs, but these six orders contained nearly 75% of the total PTUs. From these, the orders *Enterobacterales*, *Bacillales*, and *Lactobacillales* amassed more than 50% of the total PTUs, which means that most trends and conclusions drawn from the current bacterial plasmidome are likely to reflect the situation in these three orders. Besides, there are indications that not all bacterial taxa present the same trends in terms of plasmid content and propagation. Regarding host range, for example, in the order *Bacillales* only 5% of the PTUs (representing 4% of the total plasmids) exhibited Grade III or higher. In comparison, this proportion was 28% in the order *Lactobacillales* and 46% in the order *Enterobacterales*. Therefore, caution should be applied when extrapolating tendencies observed in the overall network to scarcely sampled taxa.

Once the host range of the 276 PTUs was established, we could use this information to identify possible pathways for plasmid-mediated exchanges in bacteria. This information is shown in Fig. 7a. In this figure, nodes represent bacterial species. Each time

a given PTU exhibits a host range that includes both species an edge was drawn. Thus, densely connected nodes represent bacterial species sharing numerous PTUs. An interactive version of Fig. 7a can be accessed at https://castillo.dicom.unican.es/PlasmidID/host-PTU/. Within the entire pathway map, two exchange communities stand out by the complexity of their connections, and the variety of species involved (Fig. 7b, c). The first exchange community includes all relevant enterobacterial species, connected to other relevant γ-*proteobacteria* from the orders *Pseudomonadales* and *Xanthomonadales* via *Burkholderia*, *Acidovorax* and other β-*proteobacteria*. This exchange community, which comprises several orders from γ-and β-*proteobacteria*, and isolated species from *Corynebacteria* and α-*proteobacteria*, is dependent on PTU-P1 plasmids. When removed from the map, the community divides into its *Enterobacterales*, *Pseudomonadales*, *Xanthomonadales*, and β-*proteobacteria* subcommunities (Supplementary Fig. 6). Similarly, there is a second large exchange community linking together species from the genera *Lactobacillus*, *Enterococcus*, *Staphylococcus*, and *Streptococcus* (Fig. 7c). This community stands out because species within the same genus do not necessarily localize together in the subgraph. This is particularly significant in the genus *Enterococcus*, where species *E. faecium* along with other species localize in a subgraph populated by species from *Lactobacillus* and *Pediococcus*. In contrast, *Enterococcus faecalis* is not connected to the rest of enterococci. Instead, it sits together in a densely connected subgraph with species from the genera *Streptococcus* and *Staphylococcus*. This result suggests that, although the host phylogeny seems to be the major determinant for plasmid spread, environmental factors may also determine major routes for plasmid exchange.

## Discussion

Assessing the host range of bacterial plasmids is essential to understand the overall impact of HGT on bacterial evolution. Plasmids being essential vehicles for antibiotic resistance dissemination, their transmission routes may elicit devastating consequences on human health. Yet, this assessment is complicated by the difficulty of determining which plasmids can be considered functionally equivalent. Plasmids exhibit plastic genetic architectures, making it unclear whether they form coherent groups with similar characteristics. Bacterial chromosomes posed a similar challenge, but genomic analyses have shown that microbial species have a genetic substrate[4,40,52]. This genetic substrate, however, does not guarantee a threshold of sequence conservation, nor a fixed set of genotypic traits[3,12,53]. Rather, chromosomes from the same species group together in separate paracliques in DNA/DNA similarity networks[4,40]. DNA and phenotypic conservation within these taxa are not homogeneous, some being highly monomorphic (i.e., *Mycobacterium tuberculosis*), others showing considerable sequence and phenotypic variation (i.e., *Escherichia coli*). Results presented here demonstrate that bacterial plasmids also organize into coherent genomic clusters (PTUs), similar in concept to bacterial species[4]. As in bacterial species, some PTUs were highly homogeneous (i.e., PTU-$F_S$ plasmids from *Salmonella*), while others showed low conservation in their gene repertoire (i.e., PTU-$F_E$ plasmids from *Escherichia*). Neither replication nor mobilization functions was universally conserved, thus PTUs do not exhibit phenotypic uniformity. Despite the lack of a universal conserved genomic core, PTUs show clear ANI thresholds (Fig. 2d). As in bacterial species, ANI scores between members of the same PTU were mostly above 90% sequence identity, while scores between members of different PTUs showed less than 70% identity (Fig. 2d). For bacterial chromosomes, preferential recombination

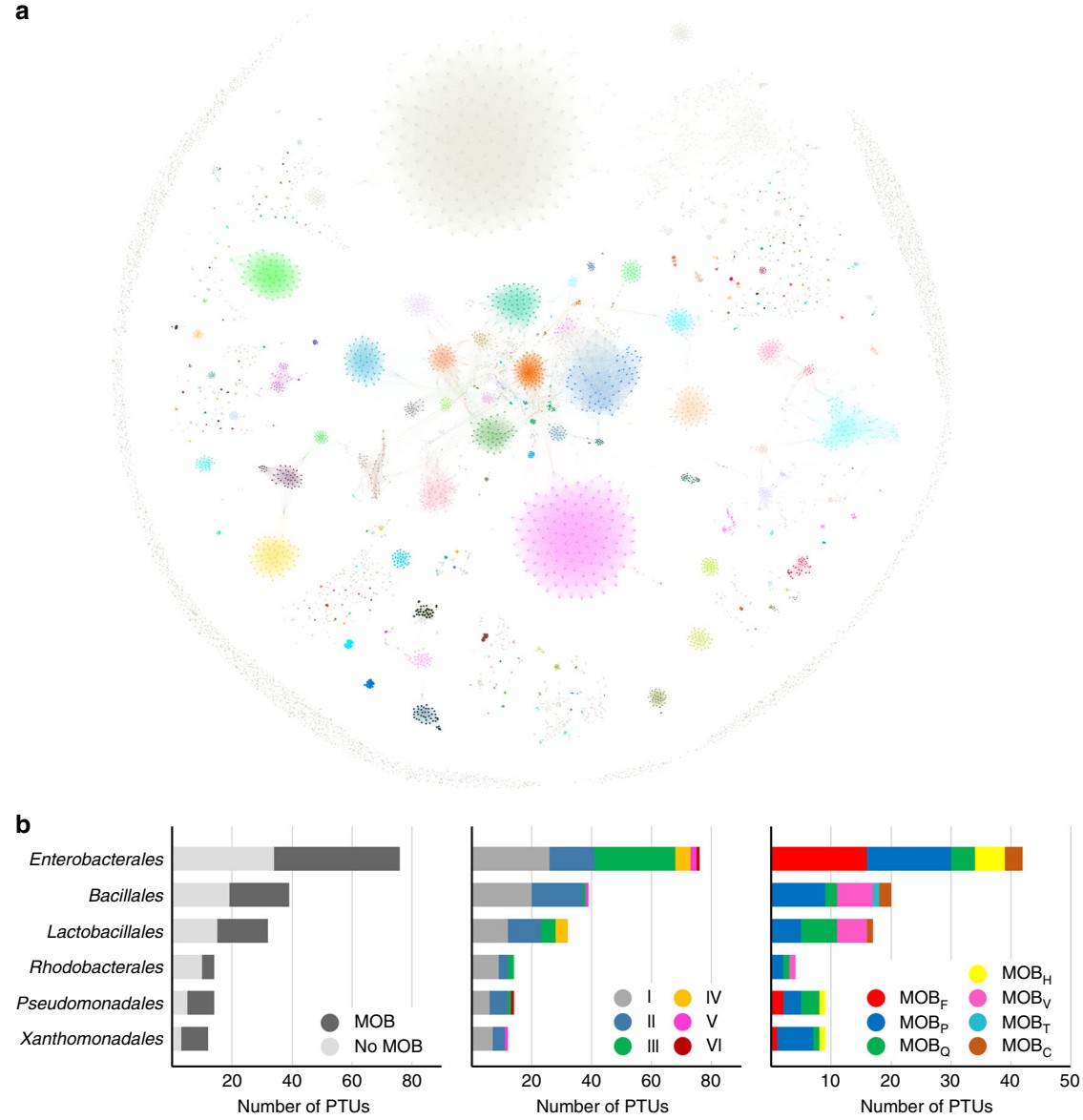

**Fig. 6 PTUs identified in the entire bacterial plasmidome. a** A colormap of the PTUs identified by PID in the $ANI_{L50}$ network of the entire bacterial plasmidome, retrieved from RefSeq84. Plasmids are colored by PTU membership. A comprehensive list of all PTUs and their members can be found in Supplementary Data 2. Source data are provided as a Source Data file. **b** Characteristics of PTUs found in the six more abundant orders. The leftmost panel indicates the proportion of MOB+ and MOB− PTUs. The panel in the middle indicates the host range distribution (Grade I–VI). The rightmost panel shows the distribution of MOB types.

between closely related genomes has been invoked as the major force maintaining genomic coherence[3,12,54,55]. Given the propensity of bacterial plasmids for recombination, it is likely that PTU identities are maintained by similar forces. A key difference is that, in plasmids, preferential recombination operates beyond the host species/genus barrier. Even broad host range PTUs of classes V and VI maintained sharp ANI boundaries, suggesting that they move enough for preferential recombination to operate across the species boundary.

Some of the PTUs identified correlated with classical Inc groups, yet ANI-based classification presents a number of noteworthy differences. First, ANI clustering allows a classification based on the entire plasmid genome, rather than a particular gene. A comparison between PTUs and Inc groups revealed that rep functions are not conserved enough to serve as a reliable universal marker. Second, ANI clustering can be applied to

plasmids from any bacterial phylum, regardless of our knowledge of the conjugation or replication mechanisms of these plasmids. Finally, statistically principled approaches, such as SBM, can be applied to evaluate the significance of the PTUs identified. Once identified, an analysis of the PTU host distribution reveals plasmid host ranges.

Although the interspecies barrier seemed permeable to most PTUs, this does not imply that plasmids can colonize all microbial taxa within their reach. Rather, proteomic and gene identity networks indicated that colonization becomes increasingly difficult as the phylogenetic distance between hosts grows. Most PTUs display a highly preferred genus host (as shown in Fig. 4c and Supplementary Fig. 5). Furthermore, the existence of a gradient of host ranges indicates that certain plasmid architectures are more versatile than others. As in higher organisms, ecological versatility correlated poorly with overall abundance. Some highly prevalent

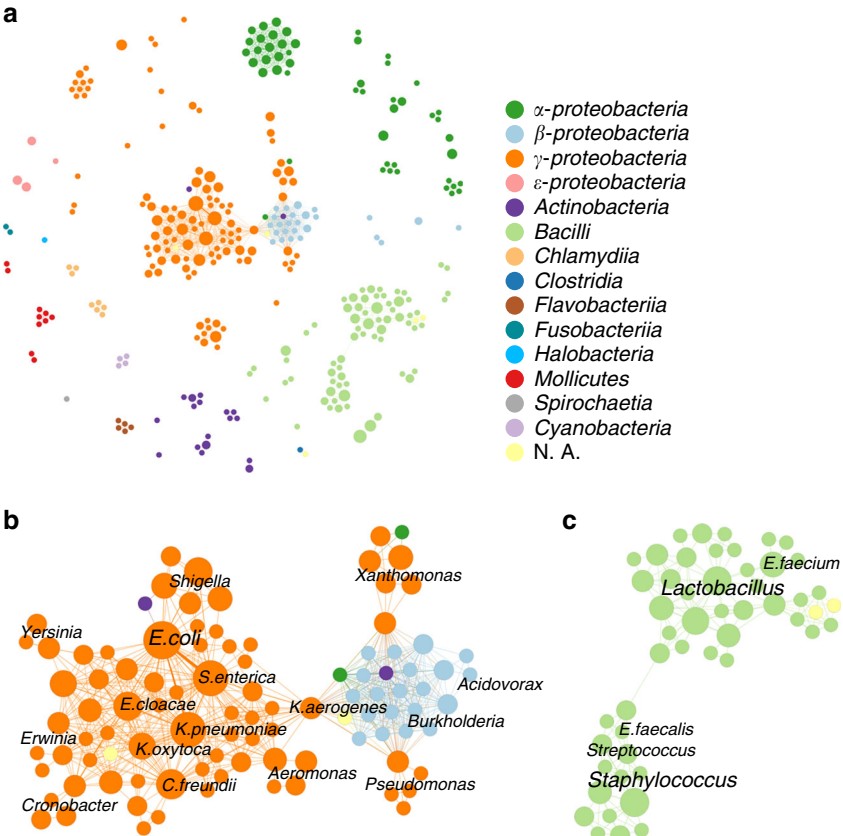

**Fig. 7 HGT pathways in bacteria drawn by PTUs. a** A network showing possible plasmid-mediated pathways for genetic exchanges between different species. Each node in the network represents a bacterial species. The size of the node is proportional to the number of PTUs identified in that species, and the color code corresponds to its cognate taxonomical class or phylum (*Cyanobacteria*). **b** The exchange community formed by species from γ and β-*proteobacteria*. As in the panel above, the size and color of the nodes represent the number of PTUs and the taxonomic class of the species considered, respectively. **c** The exchange community formed by the orders *Bacillales* and *Lactobacillales*.

PTUs, such as PTU-F$_E$, were mostly confined to a single genus, thus appearing as specialists of a given host. At the other end of the spectrum, extreme generalists such as grade VI PTU-P1 plasmids were rarely found in enterobacteria. It is yet unclear why certain PTUs exhibit broader host ranges or are more prevalent among a given taxon. Neither trait shows a clear correlation with MOB or replicon types. As observed for higher organisms, the evolution towards generalist or specialist strategies may be a way for plasmids to drive niche partition, allowing ecological coexistence. Long-term plasmid colonization requires a certain degree of plasmid/host coadaptation, even for the most promiscuous grade VI PTU-P1 plasmids[25,56]. Perhaps the ability of broad host range plasmids to colonize different species resides in their ability to generate host range shifts by means of small genetic changes[25]. Further research is required to determine whether the host range of a given PTU depends on a combination of specific phenotypic traits (replication/stability/conjugation), on its evolvability, or a combination of both.

In any case, whether a given PTU can colonize different hosts immediately, or through gradual genetic changes, is likely to have a minor impact in practical terms. Broader host range plasmids constitute hubs that spread traits across different bacterial species. For example, links between *Enterobacteriaceae*, *Pseudomonadaceae*, and *Xanthomonadaceae* were dependent on PTU-P1 plasmids, which in turn seem to be more prevalent in β-*proteobacteria*. Similarly, although there is a dense exchange network between all members of the order *Enterobacterales*, exchanges between the *Enterobacteriaceae* family (*Escherichia*,

*Klebsiella*, and *Salmonella*) and other members of the order are largely dependent on Grade V PTU-C plasmids[57]. Pathways for plasmid exchange thus seem to follow a bipartite structure, with a small number of high-grade groups being able to jump large taxonomical distances, and a larger number of low-grade plasmids mediating exchanges between closely related species. This is an important finding, because interfering with the spread of broad-range PTUs may be key to preventing the dispersal of antibiotic resistances among pathogenic species (Supplementary Table 2). It should be noted that ICEs are more frequently transferred between distant taxa[17] than plasmids. Thus, they could provide additional pathways for gene exchange. Unfortunately, not enough ICEs have been identified so far to allow their classification in taxonomic units and thus allow an extension of this work to them. Future work may further highlight the importance of ICEs in HGT and bacterial evolution.

Overall, results indicate that, although plasmid transmission is constrained by taxonomic boundaries, these are permeable enough to sustain large gene exchange networks throughout an entire bacterial order. The results presented here, however, should be considered a preliminary sketch when considering species outside the order *Enterobacterales*. Even in this highly sampled order (2535 plasmids), nearly 9% of its plasmids formed singletons. This indicates there are many more PTUs in enterobacteria besides those described here. For species outside this order, our level of knowledge is abysmally lower. For example, the entire phylum *Bacteroidetes*, prevalent members of the human flora, is represented in RefSeq84 by just 113 plasmids, nearly 90% of them

singletons. Mapping the uncharted areas of the prokaryotic plasmidome is thus a formidable, yet essential task, in order to unravel the contribution of plasmids and HGT to bacterial physiology.

## Methods

**Plasmid genome sequences and associated metadata**. Input data were retrieved from the 84th NCBI RefSeq database[58], as of 09/11/2017. It contained 10,634 putative complete plasmid genomes from bacterial and archaeal hosts. The database was manually curated, eliminating 740 sequences that corresponded to partial plasmid DNA sequences, bacterial/archaeal chromosomes, unassignable hosts, or PacBio internal control sequences. A complete list of sequences eliminated by this procedure is recorded in the last column of Supplementary Data 5. Plasmid metadata, including size, genome topology, and host were retrieved from the database annotations. Host taxonomy was categorized using the NCBI Taxonomy database[59]. The plasmid ORFeome was extracted from the corresponding GenBank files, excluding ORFs annotated as pseudogenes. In total, we retrieved 933,306 plasmid-encoded ORFs.

**Assigning MOB and Rep types**. MOB typing was based upon analysis of the plasmid relaxase sequence. Briefly, relaxases were identified from the plasmid-encoded ORFeome (933,306 proteins) by comparison to a set of Hidden Markov models (HMM) of the nine MOB types using MOBscan[50]. ORFs were classified as putative relaxases when HMM coverage was >60%, E-value was <0.01 and the independent E-value (i-E value) was <0.01. As positive controls, those proteins used to build the HMM profiles and present in the database were manually checked to match their corresponding profiles.

Replicon typing was performed using a local version of PlasmidFinder 1.3[51]. PlasmidFinder databases for *Enterobacteriaceae* and Gram-positive microorganisms were downloaded from CGE (https://cge.cbs.dtu.dk/services/PlasmidFinder/) as of 02/05/2018. Replicon typing was performed using BLAST + v-2.6.0, with >80% sequence identity and >60% length.

**Plasmid ORFeome network analysis**. To build the ORFeome networks shown in Fig. 1, we used the AcCNET bioinformatics pipeline[39]. Homologous protein clusters (HPC) were generated using kClust[60], with >30% protein identity, >80% alignment coverage and clustering E-value < 1E-14. All edges were initially assigned equal weights. The network layout was obtained using Gephi's[61] ForceAtlas2[62], a force-directed continuous algorithm[63,64] under scaling set to 1 and null edge weight influence. The Gephi network for the entire plasmid ORFeome is provided as a Source Data file.

**ANI and AF calculations**. The algorithm used for AF calculation was a gene-agnostic adaptation of Varghese's[40]. For each pairwise comparison, one of the sequences was fragmented using a sliding window of 250 bp, with a 50 bp step. All fragments were BLAST analyzed against the nonfragmented sequence, and the AF was calculated as the percentage of the windows yielding an alignment of >90% of the length with sequence identity >90% (Supplementary Fig. 2b).

ANI scores were calculated using the Ruby script from Enveomics Collection (http://enve-omics.ce.gatech.edu/enveomics/). Pairwise ANI scores were obtained by fragmenting both sequences with a sliding window of 1000 bp and 200 bp step and comparing each window against all others with BLAST. Reciprocal best hits (RBH) with sequence identity >70% and sequence alignment length >70% were further considered for ANI score calculation. ANI scores were obtained by averaging the percentage of identity of all considered windows (Supplementary Fig. 2a). The threshold for ANI analysis was set as the minimum number of windows required to cover a homologous length core, calculated as:

$$\text{length}_{\text{CORE}} = \min\left(\text{length}_{\text{P1}}, \text{length}_{\text{P2}}\right) \cdot \text{threshold}_{\text{LEN}} \quad (1)$$

In this equation, $\text{length}_P$ corresponds to the size in bp of each plasmid genome, and the threshold parameter was calculated as follows:

$$\text{threshold}_{\text{EQWIN}} = \begin{cases} 1; \textit{if } \text{length}_{\text{CORE}} \leq \text{window} \\ 1 + \left(\text{length}_{\text{CORE}} - \text{window}\right)(\text{mod})\text{step}; \text{otherwise} \end{cases} \quad (2)$$

In Eq. (2) window indicates the size of the sliding window (1000 nt) and step corresponds to the step size (200 nt).

For the AF vs ANI scatterplot shown in Fig. 2b, we used 20% as $\text{threshold}_{\text{LEN}}$ to get a broad range of behaviors of their AF and ANI interactions. Self-comparisons were not included in ANI calculations.

**Plasmidome similarity network construction**. ANI networks were constructed using the algorithm described above, with a length threshold of 50% ($\text{ANI}_{\text{L50}}$). Since the $\text{ANI}_{\text{L50}}$ score between two plasmids is symmetrical, computation time was reduced by skipping reciprocal comparisons. Network nodes, representing plasmids, were linked together by edges when they exhibited a non-null ANI score. To represent the network, we used Gephi under the linear-logarithmic version of

ForceAtlas2[62]. ForceAtlas2 was used under edge weight influence set to 1, initial layout with 0.01 scaling, and edge weights set to 1. From this initial layout, the parameter scaling was gradually incremented until the canvas limit was reached. This maximized the area occupied by the entire network, improving the visualization of different cliques and subnetworks. At this point, edge weights were recalculated to improve the resolution of paracliques. Edge weights were first adjusted according to their $\text{ANI}_{\text{L50}}$ score:

$$\text{edgeweight} = \frac{1}{1 + 20(1 - \text{ANI}/100)} \quad (3)$$

And when cliques formed, they were reverted to 0.05 for better visualization. This procedure allowed the network to self-organize in a reduced time, allowing easier visualization of plasmid relationships. From the overall Prokaryotic network, subnetworks were obtained by using Gephi's filtering capabilities. This way, the *Enterobacterales* networks shown in Fig. 3 were obtained by filtering out plasmids isolated from hosts outside this order. Gephi networks are provided as Source Data files. Summary tables and Chord diagrams were assembled from this dataset, the later ones using Circos v0.69[65].

**Plasmid clustering by density measurements**. Plasmids were manually assigned to a given ANI cluster whenever they exhibited a majority of connections to members of that particular cluster. To assess the quality of paracliques formed this way, intra and intercluster density was calculated as defined by Fortunato[66]. Briefly, the intracluster density of the subgraph is the ratio between the number of internal edges of C and the number of all possible internal edges. Similarly, the intercluster density $\delta_{\text{ext}}(C)$ is the ratio between the number of edges running from the nodes of C to the rest of the graph and the maximum number of intercluster edges possible:

$$\delta_{\text{int}}(C) = \frac{\# \text{ internal edges of } C}{n_c(n_c - 1)/2} \quad (4)$$

$$\delta_{\text{ext}}(C) = \frac{\# \text{ intercluster edges of } C}{n_c(n - n_c)} \quad (5)$$

Supplementary Data 3 lists the densities resulting from the most representative plasmid clusters thus defined in *Enterobacterales*.

**Topological clustering by PID**. In order to identify PTUs on the ANI network, we implemented a MATLAB algorithm able to separate a graph on its connected components. First, an undirected graph G is generated upon the $\text{ANI}_{\text{L50}}$ matrix. Then, the set of connected components, $G_i$ of the graph is identified using Tarjan's algorithm, a depth-first search that visits all nodes identifying strongly connected components. Since in our case G is undirected, all connections in our graph are strong. Once $G_i$ components have been identified, the algorithm separates them in a two-step fashion. First, completely connected components (those without connections to other subgraphs) are assigned to a PTU and removed from the analysis. Then, for incompletely connected components, nodes with the minimum number of connecting edges are identified and removed. The algorithm proceeds recursively until all components in the graph are completely connected. Results presented in this work have been obtained by defining PTUs as connected components of four or more members.

**Stochastic blockmodeling for community detection**. SBM algorithms were implemented in a Python environment using graph-tools[67]. Four different algorithms were applied to our ANI graphs: flat SBM, degree-corrected SBM (DC-SBM), nested SBM (NSBM) and degree-corrected hierarchical SBM (DC-NSBM). Each algorithm was initialized 100 times, and that with a lower entropy was selected and further refined using 120,000 iterations. To identify the optimal conditions for PTU identification, a series of simulated ANI networks were employed, as described in Supplementary Methods. After optimization, NSBM was chosen as the algorithm of reference. Post-NSBM we applied an a posteriori criterion that consisted in avoiding PTUs containing plasmids with no overall ANI similarity, as described in Supplementary Methods.

**Reporting summary**. Further information on research design is available in the Nature Research Reporting Summary linked to this article.

## Data availability

The sequences conforming the analyzed dataset are available in the NCBI's RefSeq repository (https://ftp.ncbi.nlm.nih.gov/refseq/release/plasmid/). The accession numbers of the sequences are listed in the Supplementary Data 5. Taxonomy data was downloaded from NCBI's Taxonomy database on Nov 24, 2017. Source data are provided with this paper.

## Code availability

Scripts required for performing the analyses described during this study can be found in https://github.com/santirdnd/ptu_paper/.

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

## Acknowledgements

This work was funded by grant BFU2017-86378-P from the Spanish MINECO.

## Author contributions

S.R.S., R.F.L., R.R., L.V., M.G.B., and M.d.T. performed the computational analysis. S.R.S., R.F.L., E.R., and F.d.C. analyzed the data. R.F.L., E.R., and F.d.l.C. wrote the manuscript.

## Competing interests

The authors declare no competing interests.
