## [Peer Review File · Nature Communications]

Reviewers' comments:

Reviewer #1 (Remarks to the Author):

The article by Redondo-Salvo et al., describes a computational comparison of all available plasmids from isolated bacteria in order to understand the level of nucleotide sequence (ANI for average nucleotide identity) and gene content (AF for alignable fraction) similarity among them and devise a bioinformatics framework to classify plasmids, following a similar effort and methods used by Varghese et al 2015 for whole bacterial genomes. In terms of biological insights that are relevant for plasmids, there is little -if any- novelty here, e.g., we knew before this study that some plasmids show narrow host specificity due to specific molecular factors and others are more broadly shared between distant bacterial or archaeal genomes, as the authors also admit in their paper. This paper seems to add to this knowledge by offering, in addition, some quantitatively insights into the frequency of each type of plasmids (in terms of host specificity), which has some value but it is not changing current knowledge. The most interesting and novel aspect of this work is probably that plasmids, like their host bacterial genomes form 95%-ANI clusters, i.e., higher sequence identity between members of a cluster vs. between clusters, and this might represent a robust way to type and classify plasmids. I find this to be novel and broadly interesting enough. For instance, maybe it was somehow expected that plasmids will show a similar picture in terms of ANI clusters with their host genomes but one could argue that this is not necessarily true since plasmids probably are under different selection pressures and move frequently between genomes (plasmid transfer).

However, I have some technical concerns that the authors should examine more carefully before more robust conclusions can emerge. Specifically, it is not clear from figure 2 that there is a correlation between AF and ANI. Figure S3 is more informative with this respect but it does look that the correlation is different for different types of plasmids based on their length and mobility. Can the authors provide the regression line between AF and ANI and based on it try to define what fraction of sequence is shared as a minimum between plasmid that show >95% ANI? If this number is very small, e.g., <10% of the genome, then it becomes challenging to type plasmids since too short sequence is alignable for calculating ANI reliably. In other words, an important result could be what is the minimum alignable fraction that we can trust to compare and classify plasmids and which cases can be typed reliably with this respect. Also, it would be important to see a figure like Fig. 3 of Jain et al., Nat. Communications, in order to clearly see that 95% is a reliable threshold for grouping plasmids. Based on Figure S3 it does seem that this may be indeed the case for many but not for all plasmid types probably (esp. small plasmids and non mobile?). Further, it is not clear if all plasmids were included in the data of Fig. S3 or some were left out based on some threshold. Because this part represents a very essential component of the manuscript, I believe more emphasis should be given to it, especially to describe clearly the cases that the proposed ANI standards do not work well. Maybe a combination of criteria can be used to account for all cases that integrate ANI and AF with hallmark genes of the plasmids for a more comprehensive approach to plasmids typing?

The paper is well written overall, with very few typos or grammatical inaccuracies and thus, requires minimum editorial editing. The Discussion section however reads more like repeating Results in several places and not real discussion of the findings, and thus, could be substantially shortened. Please see specific comments below.

Ln 85-90. Ending of Introduction section is similar to abstract. Could be a little different, e.g., give the characteristics of pTUs.

Ln 107. Taxonomic names should be italicized. Here and elsewhere. Next line seems to have an incomplete word ("contr").

Ln151-154. I believe this is novel enough. But I am not sure the author are describing figure 2 comprehensively, e.g. there are many points that show >97% ANI and AF is <50 or <20%, like a gradient, so clear separation. What is the regression line in panel B really?

Ln 203. define MOB on first usage?

Ln 356. E. faecium (E is missing for genus name).

Ln 411-413. Is it possible that we just undersample? This possibility should be discussed in my view. 2nd paragraph represents good Discussion of results; the 1st paragraph was not.

Reviewer #2 (Remarks to the Author):

In this manuscript, Redondo-Salvo et al. adopted an innovative, network approach to systematically assess the evolution of >10000 plasmid genomes in prokaryotes based on sequence similarity. The results based on the identified taxonomic units reveal a strong correlation to known archetypes, and to the inferred types based on relaxases and replicons. Based on the spectrum of host ranges of these plasmids and their mobility, the authors derived a new grading system. The authors also inferred horizontal gene transfer based on these networks. The system appears reasonable and the conclusions drawn are largely supported by the data and the results, although I cannot comment on the biological relevance of this classification and the overall study. I largely assessed the methodology of this work.

The manuscript is well-written, albeit lengthy. The methodology appears scientifically sound. The presented networks are necessarily complex, but the authors did a great job in summarising the key findings – I especially like the information presented in Figure 4. The inference of plasmid-mediated horizontal gene transfer (Figure 7) is also useful. I wish that all figures (including the supplementary figures) are in higher resolution. The 36% of plasmid singletons is remarkable, highlighting that there's still so much we don't know!

I only viewed some of the networks on the designated URL (https://castillo.dicom.unican.es/ani_web/), but some large networks did take a long time to load. A good alternative is perhaps to use the D3 library (<https://d3js.org/>) for dynamic display, e.g. previously done on thousands of prokaryote genomes <https://doi.org/10.1128/mSystems.00257-18>

I only have minor comments, as I outlined below.

Line 108: "This was in sharp contr" – incomplete sentence?

Line 133: what is this "frontier effect"?

Lines 148-149: "negligible DNA similarity" remains vague. It would be clear if this is quantified e.g. based on a threshold of percent identify (% identity <= X%).

Line 153: ">95% ANI scores over <10% of the AF" – this statement is a little counter-intuitive. Perhaps replace "over" with "covering" or "in regions covering"?

Line 159: "finely tune" -> "fine-tune"? Fine-tune the ANI thresholds to do/achieve what?

Line 186: by "genomic space", do you mean "genome-sequence space"? I interpreted "genomic space" as the actual space where the plasmid genomes are located, which is probably not what

you meant.

Line 203: the "MOB type" should be defined at first use. Also in Line 231 – it is not immediately clear at "relaxase typing" = "MOB typing"; replicon typing should also be defined at first use. These terms are not common to the broad audience (e.g. compared to MLST).

Line 205: "members are more alike between them than with individuals outside the group"... did you mean that members within a cluster are more similar to each another than to individuals external to the cluster? It is unclear what if "group" = "cluster" in this context.

Line 211: "Many of the pTUs thus defined" -> "Many of the defined pTUs", or simpler, "Many pTUs"? It's defined in the previous sentence.

Line 216: denote "MLST" here – I believe that "pMSLT" in Line 223 should be "pMLST"?

Line 273: "Bacteria" is a kingdom.

Line 503: it may be worth using the modulus symbol in writing (mod) rather than the percentage sign (commonly used in scripting/programming).

Line 555: pTid is only used once in the manuscript, thus the acronym may not be necessary.

Reviewer #3 (Remarks to the Author):

- Manuscript background information

The objective of the study is clear and defined. Analyzing plasmid genomic composition and pairwise sequence identity, authors built an atlas of the prokaryotic plasmidome. Within this map, plasmids were organized into genomic clusters defined as plasmid taxonomic units (pTUs). 261 pTUs were identified and organized into a six-grade scale (I-VI), ranging from plasmids restricted to a single host species (grade I) to plasmids able to colonize species from different Phyla (grade VI).

1-Manuscript style and content

The MS is well written and clear. However, the first two chapters are describing the process used for sequence selection and classification. This text can be heavy for readers that are not expert of bioinformatic and I suggest transferring part of this exclusion and selection process description in methods, providing a shorter text of the setting up of the process. In this context, Figures 2 and 3 are not supporting results but methods and can be supplementary material. There is also difficulty to understand the difference between Figure 3 and the following Figure 4. It can be better explained.

Repetitions are present in the discussion. Examples are the IncC and IncP host range largely discussed in results but fits well in discussion, where now it representing a repetition.

2- Specific comments

The robustness of the Atlas appeared evident when data obtained were compared to the current knowledge available on archetypal plasmids. Some information described in this study is already well established and is not the product of the analysis performed in the study itself. MOB classification has been adopted on the Atlas to identify the distribution of some pTUs into a single bacterial species, and over higher taxonomic ranks. But the verification that pTUs identified in these maps were correct, came when pTUs of Enterobacteriales fitted with the corresponding classical Inc groups. Historically, thousands of plasmids mostly identified in clinical isolates of the

order Enterobacteriales were classified in Inc groups in the last decades, thanks to molecular methods and to the collection of sequenced plasmids largely represented in the study. The information regarding archetypal plasmids and for instance their host range has been used to verify the Atlas on one side, but is also a confirmation of the previous knowledge deriving from this Atlas. The Atlas reconfirmed the validity of information available on specific Inc groups through a wider and systematic approach. Confirmatory data are important in science, but it would have been expectable to describe point by point what was confirmed by previous knowledge, representing the verification of the validity of the methods. This appears clear when the analysis is not supported by archetypal plasmids for instance for Gram-positives. In this case, the application of MOB to pTU does not improve the identification of novel archetypal plasmids. It remains as classification of diversity without further novel information on the unknown plasmid groups. Specifically, amendable sentences are listed below.

-Text at lines 211-227 represents a confirmation of the atlas based on previous knowledge on the Inc groups. MOBs are used to color the Atlas on Figure 4A and 4B but the relationship between Inc and MOB is understandable only looking carefully figure 4C, where the relationship between MOB and Inc is explicated.

It is impossible to identify the correlation among Inc and MOBs along the text. Thus, or the text is elaborated referred only to MOB families, or this relationship is clearly translated in a larger Figure. This is relevant since the text is currently referring very largely to Inc groups, but Atlas plotted only MOB superfamilies. MOB classification is highly sensitive but not specific and merged many different pTUs in one MOB group. MOB types and subtypes hierarchically represent genera and families or even orders in which plasmids can be classified and grouped together several Inc types. Thus, the presence of plasmids in different domains can be influenced by the choice to plot for MOB groups., instead of Inc groups An atlas plotting for Inc could be produced showing the effect of higher discrimination of plasmids in Inc groups.

-Lines 228-230: Given the lack of a reported group in the literature, the remaining groups were named IncE (from Enterobacteriales) followed by a number (IncE1-28).

Comment on this classification. From Figure 4 some of these IncE corresponded to previously nomenclated ColE-like plasmids. I suggest to revise the classification, recognizing pTU for ColE-like plasmids, letting the IncE1-n classification for those that were not previously known.

-In Table S2 reporting the Most abundant clusters (connected to >5% of the plasmidome) with transposase there is the >WP_088546682.1 incFII family plasmid replication initiator RepA. What does it mean? Was the replicon of the most abounding family taken out from the analysis? This should be re-checked because some of the IncE 1-n groups could be represented by the IncFII.

-Lines 234-236: Multireplicon status is normally well known on plasmids of the Enterobacteriaceae. The sentence "These results thus indicate that, while the association of pTUs to a particular replicon type is frequent, plasmids maintain their genomic identity despite shuffling replication machineries. Furthermore, replicon types are also not exclusive of a given pTU, since the same replication formula could be found in different pTUs." should be clarified. It doesn't fit with previous knowledge and what appeared from Figure 4.

-Lines 261-275. This is one of the confirmatory evidences that should reference to previous work. It is well known that IncA/C plasmids (here classified as grade V) jumped the Enterobacteriales order barrier, being identified also in *Aeromonas*, *Pseudomonas* and *Kluyvera*. As well, it has been demonstrated that IncP-plasmids are the most frequent plasmid type in *Pseudomonas* and also sporadically can be found in Enterobacteriales, where did not significantly contributed to antimicrobial resistance.

The Atlas respected the previous knowledge, but this should be commented and referenced. Figure 4B is not mentioned in the text, while it is supporting some of the conclusions

-Lines 225-227, 305-310 and along the text. The large prevalence of IncF plasmids is very well known from previous literature and the division of IncF per bacterial species (IncFy for yersinia, IncFk for klebsiella, IncFS for Salmonella) has been proposed in 2012 and is currently available for public consultation in the pMLST pages of the Oxford University; it cannot be introduced with "we named".

Reviewer #4 (Remarks to the Author):

This manuscript presents an analysis of the bipartite network of plasmid genomes and their corresponding homologous protein clusters.

Most of the analysis revolves around the identification of cohesive clusters in this network, done at first via visual inspection of the output of a force-directed graph layout algorithm (ForceAtlas2) and then via a custom algorithm based on the separation of connected components.

These approaches of identifying modular structure in networks is rudimentary, considering the state of the art. There are many pitfalls in the detection of modules in networks, such as the inability of resolving small-scale structures and (very importantly) the tendency of heuristic approaches of finding seemingly clear modules when they don't exist, e.g. in completely random networks.

The authors should familiarize themselves with the field of "community detection". A good starting point is the recent user guide by Fortunato and Hric:

Community detection in networks: A user guide
Santo Fortunato and Darko Hric
Physics Reports, Volume 659, 2016
<https://doi.org/10.1016/j.physrep.2016.09.002>

In particular the authors should consider approaches based on statistical inference, due to their principled nature and strong statistical guarantees:

Bayesian stochastic blockmodeling
Tiago P. Peixoto
<https://arxiv.org/abs/1705.10225>

There are many of-the-shelf software that employ these methods on network data, such as the graph-tool package:

<https://graph-tool.skewed.de/static/doc/demos/inference/inference.html>

Reviewer #1

The article by Redondo-Salvo et al., describes a computational comparison of all available plasmids from isolated bacteria in order to understand the level of nucleotide sequence (ANI for average nucleotide identity) and gene content (AF for alignable fraction) similarity among them and devise a bioinformatics framework to classify plasmids, following a similar effort and methods used by Varghese et al 2015 for whole bacterial genomes. In terms of biological insights that are relevant for plasmids, there is little -if any- novelty here, e.g., we knew before this study that some plasmids show narrow host specificity due to specific molecular factors and others are more broadly shared between distant bacterial or archaeal genomes, as the authors also admit in their paper. This paper seems to add to this knowledge by offering, in addition, some quantitative insights into the frequency of each type of plasmids (in terms of host specificity), which has some value but it is not changing current knowledge. The most interesting and novel aspect of this work is probably that plasmids, like their host bacterial genomes form 95%-ANI clusters, i.e., higher sequence identity between members of a cluster vs. between clusters, and this might represent a robust way to type and classify plasmids. I find this to be novel and broadly interesting enough. For instance, maybe it was somehow expected that plasmids will show a similar picture in terms of ANI clusters with their host genomes but one could argue that this is not necessarily true since plasmids probably are under different selection pressures and move frequently between genomes (plasmid transfer).

We agree with the referee that the most consequential finding of our paper is that plasmids, as their hosts, cluster into coherent genomic groups. To better convey this idea, we rewrote the abstract (lines 31-34) and the discussion of the manuscript (lines 391-395). We kept, however, the section on the issue of plasmid host range. Although we agree with the referee that it was previously known that certain plasmids have broader host ranges than others, we believe our work contains three fundamental new messages worthy of detailed explanation.

-First, our work offers the first method to infer the host range of a given PTU (rather than an individual plasmid) based on genomic sequence alone. To date, the host range of a given plasmid (not a PTU) had to be assessed either experimentally, by measuring its spread and persistence after introducing it into different strains/communities, or by inferring it from nucleotide signatures of reference genomes. Results obtained for a particular plasmid could not be extrapolated to “similar” plasmids, since there was no rigorous way of establishing systematic similarity cutoffs. By determining the host range of plasmid molecular species (PTUs), we are now able to ascribe a given plasmid a potential host range.

-Second, previous work divided plasmids into two general categories (broad and narrow host range). Our work indicates that this binary categorization is suboptimal, and offers a scale of promiscuity based on the empirical distribution of plasmids in the wild.

-Third, we focused the functional characterization on plasmid host range in the paper, since our results were rather surprising (at least for me, having been working in plasmid biology for the last 40+ years). I expected plasmids to show a widely broader host range. As it is known from many years, plasmids belonging to the same incompatibility

group were known to transfer freely and efficiently to many different species (for instance within the order Enterobacterales). Contrary to these expectations, plasmid PTUs, in general, show a much narrower effective host range, suggesting that plasmid colonization of new species (or families) is subjected to more constraints than anticipated.

We fundamentally agree with the referee in that these three aspects are a consequence of the fact that plasmids cluster into molecular species, which is the most fundamental finding of our work. However, for applied and clinical microbiologist, it is the catalog of molecular species and their distribution the aspect that is likely to have a broader impact. We hope the rewording of the abstract and discussion further clarifies this point and offers now a more balanced description of these two important aspects.

However, I have some technical concerns that the authors should examine more carefully before more robust conclusions can emerge. Specifically, it is not clear from figure 2 that there is a correlation between AF and ANI. Figure S3 is more informative with this respect but it does look that the correlation is different for different types of plasmids based on their length and mobility. Can the authors provide the regression line between AF and ANI and based on it try to define what fraction of sequence is shared as a minimum between plasmid that show >95% ANI? If this number is very small, e.g., <10% of the genome, then it becomes challenging to type plasmids since too short sequence is alignable for calculating ANI reliably. In other words, an important result could be what is the minimum alignable fraction that we can trust to compare and classify plasmids and which cases can be typed reliably with this respect. Also, it would be important to see a figure like Fig. 3 of Jain et al., Nat. Communications, in order to clearly see that 95% is a reliable threshold for grouping plasmids. Based on Figure S3 it does seem that this may be indeed the case for many but not for all plasmid types probably (esp. small plasmids and non mobile?). Further, it is not clear if all plasmids were included in the data of Fig. S3 or some were left out based on some threshold. Because this part represents a very essential component of the manuscript, I believe more emphasis should be given to it, especially to describe clearly the cases that the proposed ANI standards do not work well. Maybe a combination of criteria can be used to account for all cases that integrate ANI and AF with hallmark genes of the plasmids for a more comprehensive approach to plasmid typing?

Perhaps we did not explain sufficiently how we developed plasmid-ANI. Since plasmids contain small genomes, mobile genetic elements perturb an analysis that uses only the sequence similarity approach used by ANI, as performed for bacterial chromosomes. Therefore, we developed plasmid-ANI, which puts an additional constraint to ANI values: we added a rule by which there is a threshold of 50% of the total length of the smaller genome of the pairs that are being compared to assign an ANI value. If this threshold was not achieved, plasmid-ANI is given a value of 0. This is, precisely, what was discussed in Figure 2: the figure shows how the presence of transposons and, to a minor extent, conserved conjugation transfer systems, affect the ANI result. Setting the minimal AF value (calculated indirectly as explained below in more detail) to 50% solves this issue.

The reason why resides in the ANI algorithm (supplementary figure 2). Since it was designed for chromosomes, the ANI calculation takes into account only those segments of the comparison that show values above a threshold identity between them. Regions

with identity values below the threshold are “invisible” to the algorithm. For instance, if ANI were computed without an AF threshold, two plasmids of 100 Kb genomes sharing no sequence similarity but for a small 3 Kb transposon of 100% sequence identity, will yield an ANI score of 100%. To avoid this, we modified the ANI algorithm to include a threshold on the minimum length of the fragments with detectable homology (the alignment fraction, or AF). If the minimum AF is not satisfied, the algorithm returns ANI=0. This is usually not a problem when comparing chromosomes. Even if there are transposons, insertion sequences etc., with high sequence identity between two individual chromosomes, the overall ANI score reflects backbone conservation, since a) there is always a conserved genomic backbone between any two bacterial chromosomes and b) mobile genetic elements represent a small percentage of the length of the entire chromosomes being compared. None of these aspects is granted for plasmid, unfortunately. Thus, we need some evaluation of the length of the genome being compared. As the referee suggests, the AF would be the most obvious choice. However, AF calculations are computationally very intensive. To circumvent this problem, we compare ANI and AF in a small subset of the entire plasmidome (plasmids from enterobacterales). From this comparison, we obtain a minimum length threshold for the ANI algorithm that discards artifactual high ANI scores obtained due to transposons, insertion sequences etc. To better convey this idea in the revised manuscript, we have:

- a) Changed the naming of ANI₂₀ to ANI_{L20} to better indicate that the 20% threshold is on the length of the genome being compared and not the sequence identity.
- b) Included, in Figure 2, panels showing comparisons of AF vs ANI_{L20}, and AF vs ANI_{L50}, to show that ANI_{L50} successfully eliminates the problem of short regions with high sequence identity.
- c) As suggested by the referee, we have included a panel (Figure 2D) in the spirit of Fig 3. of Jain et al, showing that most ANI comparisons between members of the same group yield high sequence identity scores, while comparisons between members of different groups mostly produce ANI_{L50}=0 (that is, roughly less than ANI=70% for >50% of the smaller genome of each pair).

The paper is well written overall, with very few typos or grammatical inaccuracies and thus, requires minimum editorial editing. The Discussion section however reads more like repeating Results in several places and not real discussion of the findings, and thus, could be substantially shortened. Please see specific comments below.

Ln 85-90. Ending of Introduction section is similar to abstract. Could be a little different, e.g., give the characteristics of pTUs.

Agreed, the end of the introduction has been re-written.

Ln 107. Taxonomic names should be italicized. Here and elsewhere. Next line seems to have an incomplete word (“contr”).

Agreed and corrected.

Ln151-154. I believe this is novel enough. But I am not sure the author are describing figure 2 comprehensively, e.g. there are many points that show >97% ANI and AF is <50 or <20%, like a gradient, so clear separation. What is the regression line in panel B really?

We have added two new panels to Figure 2 and expanded the description in the caption of the figure. As described in our response to major comments, there is no linear correlation between ANI and AF algorithms (Supplementary Figure S2), but it depends on the length threshold included in the ANI calculation. We include AF vs ANI_{L20} and AF vs ANI_{L50} in Figure 2 to better convey this message.

Ln 203. define MOB on first usage?

Agreed, we introduced a sentence to define MOB.

Ln 356. *E. faecium* (E is missing for genus name).

Corrected

Ln 411-413. Is it possible that we just undersample? This possibility should be discussed in my view. 2nd paragraph represents good Discussion of results; the 1st paragraph was not.

While undersampling is definitively an issue for important bacterial phyla (e.g. *Bacteroidetes*), this is likely a minor problem for species from Enterobacteriaceae. This family is, comparatively, very well sampled, with more than 3K plasmids. Thus, host ranges for most enterobacterial PTUs are probably well defined, especially for those highly prevalent ones like the IncF described in the text.

Reviewer #2

In this manuscript, Redondo-Salvo et al. adopted an innovative, network approach to systematically assess the evolution of >10000 plasmid genomes in prokaryotes based on sequence similarity. The results based on the identified taxonomic units reveal a strong correlation to known archetypes, and to the inferred types based on relaxases and replicons. Based on the spectrum of host ranges of these plasmids and their mobility, the authors derived a new grading system. The authors also inferred horizontal gene transfer based on these networks. The system appears reasonable and the conclusions drawn are largely supported by the data and the results, although I cannot comment on the biological relevance of this classification and the overall study. I largely assessed the methodology of this work.

The manuscript is well-written, albeit lengthy. The methodology appears scientifically sound. The presented networks are necessarily complex, but the authors did a great job in summarising the key findings – I especially like the information presented in Figure 4. The inference of plasmid-mediated horizontal gene transfer (Figure 7) is also useful. I wish that all figures (including the supplementary figures) are in higher resolution. The 36% of plasmid singletons is remarkable, highlighting that there's still so much we don't know!

I only viewed some of the networks on the designated URL (https://castillo.dicom.unican.es/ani_web/), but some large networks did take a long time to load. A good alternative is perhaps to use the D3 library (<https://d3js.org/>) for dynamic display, e.g. previously done on thousands of prokaryote genomes <https://doi.org/10.1128/mSystems.00257-18>

We are sorry for the lack of resolution, we have now included high-resolution figures for a better visualization. Also, we have upgraded the webpage with a new format that includes data requested by the referees, and for large networks it should load in ~4 secs.

I only have minor comments, as I outlined below.

Line 108: “This was in sharp contr” – incomplete sentence?

The typo has been corrected in the main text.

Line 133: what is this “frontier effect”?

We changed the words “frontier effect” by “fact”, which is more precise.

Lines 148-149: “negligible DNA similarity” remains vague. It would be clear if this is quantified e.g. based on a threshold of percent identify (% identity \leq X%).

OK. Instead of “negligible” we write now “below 70% identity”

Line 153: “>95% ANI scores over <10% of the AF” – this statement is a little counter-intuitive. Perhaps replace “over” with “covering” or “in regions covering”?

OK. Following a request from referee #1, we use ANI_{L20} (or ANI_{L50}) instead of ANI, for added clarity. Besides, we change the sentence as suggested by this referee, substituting “in regions covering” for “over” in the given sentence.

Line 159: “finely tune” → “fine-tune”? Fine-tune the ANI thresholds to do/achieve what?

OK. This sentence has disappeared in the revised text.

Line 186: by “genomic space”, do you mean “genome-sequence space”? I interpreted “genomic space” as the actual space where the plasmid genomes are located, which is probably not what you meant.

OK, the sentence is changed as requested.

Line 203: the “MOB type” should be defined at first use. Also in Line 231 – it is not immediately clear at “relaxase typing” = “MOB typing”; replicon typing should also be defined at first use. These terms are not common to the broad audience (e.g. compared to MLST).

OK. The terms “MOB type” and “relaxase” are now defined in the sentences following the old line 203.

Line 205: “members are more alike between them than with individuals outside the group”... did you mean that members within a cluster are more similar to each other than to individuals external to the cluster? It is unclear what if “group” = “cluster” in this context.

OK. The term “cluster” substitutes for “group”.

Line 211: “Many of the pTUs thus defined” → “Many of the defined pTUs”, or simpler, “Many pTUs”? It’s defined in the previous sentence.

OK. The term “thus defined” is eliminated from the sentence.

Line 216: denote “MLST” here – I believe that “pMSLT” in Line 223 should be “pMLST”?

OK. Typo corrected.

Line 273: “Bacteria” is a kingdom.

OK, corrected.

Line 503: it may be worth using the modulus symbol in writing (mod) rather than the percentage sign (commonly used in scripting/programming).

OK, corrected.

Line 555: pTId is only used once in the manuscript, thus the acronym may not be necessary.

OK. The acronym pTId was used by mistake. We used PID through the text, so the acronym was changed in that title.

Reviewer #3

- Manuscript background information

The objective of the study is clear and defined. Analyzing plasmid genomic composition and pairwise sequence identity, authors built an atlas of the prokaryotic plasmidome. Within this map, plasmids were organized into genomic clusters defined as plasmid taxonomic units (pTUs). 261 pTUs were identified and organized into a six-grade scale (I-VI), ranging from plasmids restricted to a single host species (grade I) to plasmids able to colonize species from different Phyla (grade VI).

1-Manuscript style and content

The MS is well written and clear. However, the first two chapters are describing the process used for sequence selection and classification. This text can be heavy for readers that are not expert of bioinformatic and I suggest transferring part of this exclusion and selection process description in methods, providing a shorter text of the setting up of the process. In this context, Figures 2 and 3 are not supporting results but methods and can be supplementary material. There is also difficulty to understand the difference between Figure 3 and the following Figure 4. It can be better explained.

Repetitions are present in the discussion. Examples are the IncC and IncP host range largely discussed in results but fits well in discussion, where now it representing a repetition.

While we agree with the referee in that certain readers may find the description of the methods too heavy, we had to achieve a compromise between this request and those of refs #1 and #4, who demanded further description on the technical side of the paper. Trying to achieve such balance, all new technical details regarding stochastic block models and Bayesian analysis have been moved to a supplementary methods note. We have also substantially re-written some of the results section, in an attempt to make the methods more understandable to non-experts in bioinformatic analysis.

2- Specific comments

The robustness of the Atlas appeared evident when data obtained were compared to the current knowledge available on archetypal plasmids. Some information described in this study is already well established and is not the product of the analysis performed in the study itself.

MOB classification has been adopted on the Atlas to identify the distribution of some pTUs into a single bacterial species, and over higher taxonomic ranks. But the verification that pTUs identified in these maps were correct, came when pTUs of Enterobacteriales fitted with the corresponding classical Inc groups. Historically, thousands of plasmids mostly identified in clinical isolates of the order Enterobacteriales were classified in Inc groups in the last decades, thanks to molecular methods and to the collection of sequenced plasmids largely represented in the study. The information regarding archetypal plasmids and for instance their host range has been used to verify the Atlas on one side, but is also a confirmation of the previous knowledge deriving from this Atlas. The Atlas reconfirmed the validity of information

available on specific Inc groups through a wider and systematic approach. Confirmatory data are important in science, but it would have been expectable to describe point by point what was confirmed by previous knowledge, representing the verification of the validity of the methods. This appears clear when the analysis is not supported by archetypal plasmids for instance for Gram-positives. In this case, the application of MOB to pTU does not improve the identification of novel archetypal plasmids. It remains as classification of diversity without further novel information on the unknown plasmid groups.

With due respect, we have to disagree with this interpretation. We think it is important to clarify this point, since it pertains the core of our paper. The goal of the paper is to develop a method of systematic, taxonomic classification for plasmids, based on genomic data, as those recently developed for bacterial chromosomes (Jain et al, Nat. Comm, 2018). Such method could be very useful to plasmid epidemiologists and microbiologists in general. Our method generates a classification, and this classification sometimes overlaps with those generated with previous methods (such as MOB and pMLST typing). This is to be expected, since these methods also employ partial sequence information. However, there are a number of differences between ours and previous classification schemes:

1.- The robustness of our approach is not dependent nor validated by current knowledge of archetypal plasmids, but by the convergence in the results yielded by different clustering algorithms. The clusters thus defined sometimes coincide with previous Inc groups, but this is not guaranteed nor universal. Some “classical” Inc groups are of higher taxonomic rank than PTUs, while others are of lower rank. Besides, many groups (53 out of 83 within the Enterobacterales) were not previously defined by Inc types. As a consequence, we decided to drop the Inc names for the PTUs, and will retain the capitals with added numbers or suffixes when appropriate. Therefore, the groups will be named PTU-P1, etc., to clarify this point.

2. *Inc* groups are mostly defined by the sequences of their rep regions, which we find is not an appropriate classification tool (as shown in figure 4). Plasmids within the same PTU having multiple or different rep regions are the norm, rather than the exception. This is not equivalent to stating that multi-replicon plasmids are common. Our results show that plasmids *from the same PTU* sometimes contain entirely alternate replication formulas.

3. Presently, there is no plasmid classification scheme besides some proteobacteria and firmicutes, and Inc classification is not hierarchical nor taxonomic. Plasmid ANI, as shown in Figure 4, is taxonomic (as for bacterial chromosomes, see Jain et al. Nat Com. 2018). Since it depends only on sequence data, it is also universal. PTUs have the same univocal meaning in all bacterial ranks.

4. The information that we analyze, and on which we focus, is not already well established. The following is new (besides the methods themselves): (i) Plasmids are classified according to overall homology, not by the sequence of their replication regions. (ii) Our classification is taxonomic. Plasmids are classified in classes (MOB), families and PTUs. (iii) Most plasmids (82% of the RefSeq plasmids of Enterobacterales) can be unequivocally classified at once in a defined short set of PTUs, rendering PTU analysis a feasible task from here onwards. (iv) PTUs seem to have a

rather reduced host range, being largely “constrained” by the host, or adapted to it. The mechanisms for these constraints and adaptations can now be targeted experimentally (because we have defined PTUs). (v) There is not, up to now, a host range classification for PTUs, but for individual plasmids. We provide a six-grade classification of PTU host ranges.

Thanks to the referee comments we now realize that by stressing the commonalities found between our method and previous Inc typing schemes, some of the findings remained relatively obscure and confusing in the paper. To amend this problem, we have included a paragraph in the discussion stressing the differences between ANI vs Inc classifications, and as indicated before, dropped the *Inc* denomination altogether.

Specifically, amendable sentences are listed below.

-Text at lines 211-227 represents a confirmation of the atlas based on previous knowledge on the Inc groups. MOBs are used to color the Atlas on Figure 4A and 4B but the relationship between Inc and MOB is understandable only looking carefully figure 4C, where the relationship between MOB and Inc is explicated.

It is impossible to identify the correlation among Inc and MOB along the text. Thus, or the text is elaborated referred only to MOB families, or this relationship is clearly translated in a larger Figure. This is relevant since the text is currently referring very largely to Inc groups, but Atlas plotted only MOB superfamilies. MOB classification is highly sensitive but not specific and merged many different pTUs in one MOB group. MOB types and subtypes hierarchically represent genera and families or even orders in which plasmids can be classified and grouped together several Inc types. Thus, the presence of plasmids in different domains can be influenced by the choice to plot for MOB groups., instead of Inc groups An atlas plotting for Inc could be produced showing the effect of higher discrimination of plasmids in Inc groups.

As stated above, *Inc* is not a suitable taxonomic classification approach. We reason this at large in the text, and it is strikingly shown in figure 4 in which we see that most PTUs contain more than one possible replication region. MOB defines plasmid classes, which contain several PTUs. It is important to emphasize that we did not use MOB to identify correlations: PTUs self-organize according to ANI-based DNA sequence comparisons. MOB identification and coloring and Rep typing using PlasmidFinder are performed afterwards, independently from the generation of the network and the definition of PTUs. They represent tags on the nodes, similar to host species, plasmid size or NCBI accession number. As such, they do not influence the composition of PTUs in different domains: this only depends on the number of plasmids present and their DNA sequence.

As mentioned before, to clarify this point we have dropped altogether the *Inc* denomination, in favour of PTU-XXX.

-Lines 228-230: Given the lack of a reported group in the literature, the remaining groups were named IncE (from Enterobacterales) followed by a number (IncE1-28). Comment on this classification. From Figure 4 some of these IncE corresponded to previously nomenclated ColE-like plasmids. I suggest to revise the classification, recognizing pTU for ColE-like plasmids, letting the IncE1-n classification for those that were not previously known.

ColE-like plasmids may or may not contain replication proteins. The most prevalent ColE1-like plasmids replicate using an RNA transcript of the plasmid that is used as a primer by the host polymerase I. Therefore, rep regions in ColE1-like plasmids cannot be compared to Rep-dependent plasmids, and there is no consensual definition of what constitutes a ColE1-like incompatibility group. In fact, very similar ColE1-like plasmids are compatible, although they obviously belong to the same PTU, according to DNA-comparison rules. Since we have dropped the Inc nomenclature for good in this manuscript, to avoid this type of misunderstandings, we consider it is appropriate to maintain the E1 to E53 nomenclature for the “new” PTUs.

-In Table S2 reporting the Most abundant clusters (connected to >5% of the plasmidome) with transposase there is the >WP_088546682.1 incFII family plasmid replication initiator RepA. What does it mean? Was the replicon of the most abounding family taken out from the analysis?

This should be re-checked because some of the IncE 1-n groups could be represented by the IncFII.

Table S2 contains the most abundant protein clusters present in the bacterial plasmidome, independently of their distribution in one or several PTUs. The total abundance of these clusters is thus the product of the number of plasmids containing that particular protein, rather than their categorization. The most abundant clusters are, unsurprisingly, transposases and other genes from diverse MGEs frequently found in plasmids. The reason for the high prevalence of >WP_088546682.1 is two-fold: on the one hand, it is one of the replication proteins of some of the most populated PTUs in the network (those of F-like plasmids). On the other hand, it is also present in other PTUs with little DNA similarity to PTU-F, such as MOB_p PTU-E25. These plasmid groups are highly prevalent in Enterobacteria, and due to sampling bias represent a large fraction of the entire bacterial plasmidome. We thus believe that the abundance of this cluster is mostly a consequence of the skewed sampling across bacterial phyla.

-Lines 234-236: Multireplicon status is normally well known on plasmids of the Enterobacteriaceae. The sentence “These results thus indicate that, while the association of pTUs to a particular replicon type is frequent, plasmids maintain their genomic identity despite shuffling replication machineries. Furthermore, replicon types are also not exclusive of a given pTU, since the same replication formula could be found in different pTUs.” should be clarified. It doesn’t fit with previous knowledge and what appeared from Figure 4.

As rightly pointed out by the referee, this is an important discrepancy between PTU and Inc classification. There is a column in Figure 4 that shows different shade of blue for different rep regions within each PTU. Exactly which Rep each plasmid contains is shown in Table S4. Our results indicate that Rep types and PTUs are moderately correlated. That is, most PTUs have one Rep (or one Rep formula, for multireplicon plasmids) that is usually the most frequent. Yet as shown in Figure 4 in many PTUs there is a considerable degree of variation in the replicon structure.

-Lines 261-275. This is one of the confirmatory evidences that should reference to previous work.

It is well known that IncA/C plasmids (here classified as grade V) jumped the Enterobacteriales order barrier, being identified also in *Aeromonas*, *Pseudomonas* and *Kluyvera*. As well, it has been demonstrated that IncP-plasmids are the most frequent plasmid type in *Pseudomonas* and also sporadically can be found in Enterobacteriales, where did not significantly contributed to antimicrobial resistance. The Atlas respected the previous knowledge, but this should be commented and referenced.

We agree with the referee, we now have four references on IncP-1 plasmids and their distribution (Zhang et al, 2015; Yano et al, 2013; Sen et al, 2013; Norberg et al, 2011), and we have included one review on IncC plasmids that explains previous knowledge on their host range (Ambrose et al, 2018).

Figure 4B is not mentioned in the text, while it is supporting some of the conclusions

OK. A new sentence has been added to Results (lines 242-243) to explain in the text what is in Figure 4, panels A and B.

-Lines 225-227, 305-310 and along the text. The large prevalence of IncF plasmids is very well known from previous literature and the division of IncF per bacterial species (IncFy for yersinia, IncFk for klebsiella, IncFS for Salmonella) has been proposed in 2012 and is currently available for public consultation in the pMLST pages of the Oxford University; it cannot be introduced with “we named”.

In the revised version of the manuscript we have dropped the Inc notation altogether, but we name these PTUs just by analogy to the pMLST-based classification. In old line 238 we specifically cite ref. 49 as the convention used to name PTUs.

Referee #4

These approaches of identifying modular structure in networks is rudimentary, considering the state of the art. There are many pitfalls in the detection of modules in networks, such as the inability of resolving small-scale structures and (very importantly) the tendency of heuristic approaches of finding seemingly clear modules when they don't exist, e.g. in completely random networks.

In particular the authors should consider approaches based on statistical inference, due to their principled nature and strong statistical guarantees:

Bayesian stochastic blockmodeling
Tiago P. Peixoto
<https://arxiv.org/abs/1705.10225>

There are many of-the-shelf software that employ these methods on network data, such as the graph-tool package:

<https://graph-tool.skewed.de/static/doc/demos/inference/inference.html>

We would like to thank the referee for his/her suggestions, which we believe have considerably improved the statistical support of the plasmid clusters. In our revised version of the manuscript, we employ SBMs to check for the statistical robustness of the groups found in the network.

In our initial submission, SBMs were not used. The connectivity and membership of apparent communities of our network followed power-law distributions, and we had the erroneous impression that SBMs were poorly suited for tackling heavy-tailed distributions like these. Our initial strategy was thus to identify communities based on network topology, and then validate them independently using t-stochastic network embedding (t-SNE). However, as pointed out by the referee, these methods lack statistical support for the communities thus identified, and SBMs are clearly superior in that respect. In response to the referee's comments, in this new version we dropped t-SNE validation and opted for SBMs instead.

In our revised manuscript, we use SBMs to identify PTUs, yet we had to include an important modification. Conventional SBM approaches cluster together nodes with similar connectivity patterns, since its null hypothesis is to split communities only when there is enough statistical support for doing so. While statistically sound, this is not suited to the description of our network. PTUs have an implicit genetic similarity requirement, which makes it biologically absurd to cluster in the same "molecular species" individuals with 0 genetic identity. The obvious corollary is that plasmids with no genetic similarity should not be put together in the same PTU. That is, our null hypothesis should be the opposite: for our particular application it is preferable to have an over splitting of communities rather than mixing together genetically unrelated ones.

Instead of introducing this as a prior for the SBM algorithm, we resorted to applying it as a posteriori check of the PTUs identified. We assessed the viability of this approach by simulating synthetic networks of 2500 plasmids distributed on 300 PTUs and a

gamma distribution similar to that naturally observed ($\alpha=0.3$, $\beta=1$). The procedure is more thoroughly described in a supplementary materials note. PTU definition using a naïve SBM approach scored an $NMI = 0.965967 \pm 0.00458$, while a correction to avoid joining together PTUs with 0 genetic identity yielded $NMI = 0.99674 \pm 0.00075$.

We thus opted for this method for community detection in our networks, and compared the results with those obtained using PID (community detection based on network topology). When applied to the enterobacterales network, agreement between the communities detected by both methods was 92%. Differences arose mostly in the large PTU-F cluster. This cluster includes ~200 plasmids, and it is remarkably heterogeneous in terms of internal connectivity among its members. SBM was able to subdivide it in several subgroups, but the hierarchy demonstrates the presence of this supercluster. Interestingly, this is also in agreement with previous biological knowledge, that suggested that PTU-F plasmids represent a super-group with internal structure (Fernandez-Lopez et al, 2017). To convey all this new information, we have included a new supplementary table (Supplementary table S6) that shows the statistical support of each PTU.

REVIEWERS' COMMENTS:

Reviewer #1 (Remarks to the Author):

The revised manuscript by Redondo-Salvo et al., represents a much improved version of the originally submitted manuscript. The authors have addressed all my major concerns. I have only a few minor concerns remaining as described below.

Line 53 and elsewhere. "genetic kinship" . I think this can be stated differently (more accurately and technical), e.g., higher genetic relatedness.

Lines 84-85. I think ecological cohesiveness is also possible as an alternative mechanism for the clusters. I would rephrase this sentence about recombination or just remove it; e.g., you do not need to invoke recombination here really.

Line 157. "DNA similarity". For DNA, identity is the word to use; similarity for proteins is ok. Also, it would be nice to briefly mention if L20 in ANIL20 refers to the total length of the shortest plasmid in the comparison (i.e. at least 20% of the length should be aligned for ANI calculation) or just the length of the individual genes of the plasmid compared to be at least 20% in the alignment. It is not clear as stated in the Results, and I think it is better to mention here than refer the reader to the Methods section instead.

Line 170-173. Why the difference in computing time? Perhaps a brief description of this would be useful here. Also, FastANI should be able to handle this plasmid dataset easily, albeit some tweaking of the code would be necessary to adjust it to the plasmid characteristics (e.g., shorter sequences than typical bacterial chromosomes).

Line 410. "differences noteworthy". I would reverse the order of these two words.

Reviewer #3 (Remarks to the Author):

The revised manuscript improved and it is much more clear and readable. Figures are of excellent qualities and results and conclusions are clearer.
Replies and changes done by authors satisfied my concerns.

Reviewer #4 (Remarks to the Author):

The authors have incorporated my suggestion of improving the community detection approach by using statistical inference of SBMs.

I am satisfied with the modifications.

I would only suggest for the authors to give proper attribution to methods used, by citing the relevant literature.

We wish to thank the reviewers for their helpful and constructive comments. Their suggestions have definitely helped to create an improved version of the paper which we attach here as a revision.

REVIEWERS' COMMENTS:

Reviewer #1 (Remarks to the Author):

The revised manuscript by Redondo-Salvo et al., represents a much improved version of the originally submitted manuscript. The authors have addressed all my major concerns. I have only a few minor concerns remaining as described below.

Line 53 and elsewhere. "genetic kinship" . I think this can be stated differently (more accurately and technical), e.g., higher genetic relatedness.

Changed as indicated by the referee.

Lines 84-85. I think ecological cohesiveness is also possible as an alternative mechanism for the clusters. I would rephrase this sentence about recombination or just remove it; e.g., you do not need to invoke recombination here really.

Corrected as suggested by the referee.

Line 157. "DNA similarity". For DNA, identity is the word to use; similarity for proteins is ok.

Corrected as suggested by the referee.

Also, it would be nice to briefly mention if L20 in ANIL20 refers to the total length of the shortest plasmid in the comparison (i.e. at least 20% of the length should be aligned for ANI calculation) or just the length of the individual genes of the plasmid compared to be at least 20% in the alignment. It is not clear as stated in the Results, and I think it is better to mention here than refer the reader to the Methods section instead.

The sentence recommended by the referee has been introduced in the main text.

Line 170-173. Why the difference in computing time? Perhaps a brief description of this would be useful here. Also, FastANI should be able to handle this plasmid dataset

easily, albeit some tweaking of the code would be necessary to adjust it to the plasmid characteristics (e.g., shorter sequences than typical bacterial chromosomes).

Unfortunately, the reasons behind are a bit technical and probably not suited for the general reader. Basically, the sliding windows employed by the AF algorithm move with 4x higher resolution than ANI. This results in 16x the number of pairwise comparisons.

FastANI was deployed after we started our study. Thus, we kept ANI in order to keep a unified methodology. Nevertheless, we have checked that FastANI and ANI perform similarly well in our datasets.

Line 410. "differences noteworthy". I would reverse the order of these two words.

Changed as indicated by the referee.

Reviewer #3 (Remarks to the Author):

The revised manuscript improved and it is much more clear and readable. Figures are of excellent qualities and results and conclusions are clearer.

Replies and changes done by authors satisfied my concerns.

Reviewer #4 (Remarks to the Author):

The authors have incorporated my suggestion of improving the community detection approach by using statistical inference of SBMs.

I am satisfied with the modifications.

I would only suggest for the authors to give proper attribution to methods used, by citing the relevant literature.

Original research on Stochastic Block Models from Holland et al. has been properly attributed. Additionally, a citation for graph-tool software has been added to the main text and Supplementary Information.